# Genetics and Cognitive Vulnerability to Sleep Deprivation in Healthy Subjects: Interaction of ADORA2A, TNF-α and COMT Polymorphisms

**DOI:** 10.3390/life11101110

**Published:** 2021-10-19

**Authors:** Mégane Erblang, Catherine Drogou, Danielle Gomez-Merino, Arnaud Rabat, Mathias Guillard, Pascal Van Beers, Michael Quiquempoix, Anne Boland, Jean François Deleuze, Robert Olaso, Céline Derbois, Maxime Prost, Rodolphe Dorey, Damien Léger, Claire Thomas, Mounir Chennaoui, Fabien Sauvet

**Affiliations:** 1Unité Fatigue et Vigilance, Institut de Recherche Biomédicale des Armées (IRBA), EA 7330 VIFASOM, Université de Paris, 75004 Paris, France; megane.erblang@gmail.com (M.E.); catherine.drogou@gmail.com (C.D.); dangomez51@gmail.com (D.G.-M.); arnaud.rabat.irba@gmail.com (A.R.); mathias.guillard55@gmail.com (M.G.); pavanbeers@gmail.com (P.V.B.); michael.quiquempoix@gmail.com (M.Q.); rodolphe.dorey@gmail.com (R.D.); mounirchennaoui@gmail.com (M.C.); 2EA 7330 VIFASOM, Université de Paris, APHP, Hôtel Dieu, Centre du Sommeil et de la Vigilance, 75004 Paris, France; damien.leger@aphp.fr; 3Laboratoire de Biologie de l’Exercice pour la Performance et la Santé (LBEPS), UMR, Université d’Evry, IRBA, Université de Paris Saclay, 91025 Evry-Courcouronnes, France; claire.thomas@univ-evry.fr; 4CEA, Centre National de Recherche en Génomique Humaine, Université Paris-Saclay, 91057 Evry, France; boland@cng.fr (A.B.); deleuze@cng.fr (J.F.D.); olaso@cng.fr (R.O.); derbois@cng.fr (C.D.); 5Ecole de Santé Militaire de Lyon-Bron, 331 Avenue du Général-de-Gaulle, 69500 Bron, France; maxime.prost@emslb.fr

**Keywords:** sleep deprivation, cognitive responses, genetic polymorphisms, A2A receptor, TNF-α, IL1-β, IL-6, COMT

## Abstract

Several genetic polymorphisms differentiate between healthy individuals who are more cognitively vulnerable or resistant during total sleep deprivation (TSD). Common metrics of cognitive functioning for classifying vulnerable and resilient individuals include the Psychomotor Vigilance Test (PVT), Go/noGo executive inhibition task, and subjective daytime sleepiness. We evaluated the influence of 14 single-nucleotide polymorphisms (SNPs) on cognitive responses during total sleep deprivation (continuous wakefulness for 38 h) in 47 healthy subjects (age 37.0 ± 1.1 years). SNPs selected after a literature review included SNPs of the adenosine-A2A receptor gene (including the most studied rs5751876), pro-inflammatory cytokines (TNF-α, IL1-β, IL-6), catechol-O-methyl-transferase (COMT), and PER3. Subjects performed a psychomotor vigilance test (PVT) and a Go/noGo-inhibition task, and completed the Karolinska Sleepiness Scale (KSS) every 6 h during TSD. For PVT lapses (reaction time >500 ms), an interaction between SNP and SDT (*p* < 0.05) was observed for ADORA2A (rs5751862 and rs2236624) and TNF-α (rs1800629). During TSD, carriers of the A allele for ADORA2A (rs5751862) and TNF-α were significantly more impaired for cognitive responses than their respective ancestral G/G genotypes. Carriers of the ancestral G/G genotype of ADORA2A rs5751862 were found to be very similar to the most resilient subjects for PVT lapses and Go/noGo commission errors. Carriers of the ancestral G/G genotype of COMT were close to the most vulnerable subjects. ADORA2A (rs5751862) was significantly associated with COMT (rs4680) (*p* = 0.001). In conclusion, we show that genetic polymorphisms in ADORA2A (rs5751862), TNF-α (rs1800629), and COMT (rs4680) are involved in creating profiles of high vulnerability or high resilience to sleep deprivation. (NCT03859882).

## 1. Introduction

Many occupations such as military personnel, firefighters, police officers, and hospital staff are often exposed to sleep loss, which increases the risk of accidents [1]. It has long been established that total sleep deprivation (TSD) increases daytime sleepiness and decreases sustained attention and executive cognitive processes [2,3]. Sustained attention, executive function, and subjective sleepiness are particularly important for military personnel (pilots, infantry, and sailors, who are often on external operations (OPEX)) [4]. In sleep loss studies conducted in the laboratory or workplaces, the most consistently and dramatically affected cognitive capacity is sustained attention assessed through the psychomotor vigilance task (PVT) [5,6,7]. However, there are inter-individual differences in cognitive responses to sleep deprivation [8,9] attributed to systemic inter-individual differences in sleep/wake homeostasis [10] or determined by genetic polymorphisms [11,12]. The influence of single-nucleotide polymorphisms (SNPs) on cognitive responses to TSD has been mainly observed on tasks involved in different levels of cognition, such as the psychomotor vigilance test (PVT) and the executive Go/noGo inhibition task, and also working memory, decision making, and flexibility [9]. The PVT is one of the most sensitive cognitive tests to sleep deprivation-related sleep pressure, which induces increased levels of extracellular adenosine and increased neuronal activity in a parieto-frontal network [13]. The core executive functioning (such as behavioral inhibition, working memory, and cognitive flexibility) requests optimal functioning of parieto-frontal and fronto-striatal networks, served by COMT activity, the gene of catechol-O-methyl transferase, the main enzyme degrading catecholamines, as well as dopamine levels [14].

Adenosinergic mechanisms that are implicated in sleep/wake homeostasis were previously evidenced to contribute to individual differences in waking-induced impairment of neurobehavioral performance and functional aspects of EEG topography associated with sleep deprivation [10]. In a recent study, we showed the influence of ADORA2A genetic polymorphisms on the risk of sleep complaints and insomnia in 1023 active workers [15]. In addition, several studies, including ours, have shown an association between cognitive performances and ADORA2A and ADA polymorphisms during conditions of sleep deprivation or nap sleep [16,17,18,19]. The genetic influence of the pro-inflammatory cytokine TNF-α has also been demonstrated to play a role in the occurrence of inter-individual differences in vulnerability to neurobehavioral impairments during sleep loss [20]. This was based on previous studies showing the role of TNF-α in sleep/wake regulation [21]. The TNFα polymorphism (rs1800629) has been found associated with phenotypic inter-individual differences in vulnerability to PVT performance impairment related to sleep loss [20], as well as common symptoms experienced across chronic conditions such as pain, sleep, fatigue, and cognitive symptoms [22]. Interestingly, this functional polymorphism impacts either the secretion of the protein or the level of transcription of the gene [23,24]. Our recent study showed the vulnerability of T allele carriers (compared to homozygotes C/C) of the rs5751876 of ADORA2A and A allele carriers (compared to homozygotes G/G) of TNF-α in PVT lapses performance during TSD [19]. In addition, several studies focused on the role of genetic catecholaminergic markers including COMT, the gene of catechol-O-methyl transferase, the main enzyme degrading catecholamines, and DAT1 and DRD2—the human dopaminergic transporter and receptor, and have shown association with the neurobehavioral vulnerability to sleep deprivation [25,26]. Subjects with the Val allele of COMT Val158Met (rs4680), either as heterozygotes or homozygotes, revealed a considerable vulnerability in adaptive decision making using the Go/noGo reversal learning task during TSD [25]. DAT1 and DRD2 (rs6277 also named DRD2 C957T) genotypes have been shown to distinctly modulate sleep deprivation (40 h of wakefulness)-induced changes in subjective sleepiness, PVT lapses, and TAR (the theta-to-alpha power ratio) in the waking EEG, according to inverted U-shaped relationships [26]. Carriers of the C allele of the DRD2 C957T polymorphism have been shown resilient to sleep deprivation effects on cognitive flexibility, but not vigilant attention [27]. The circadian clock gene PERIOD3 (PER3) was also found associated with individual differences in subjective sleepiness and PVT lapses during TSD [28], while the association was on the working memory n-back task rather than on PVT [29]. Finally, the BDNF Val66Met polymorphism (rs6265) has been found to modulate sleep intensity and to impact an individual’s vulnerability to sleep deprivation [30,31], with Met carriers performing more poorly on neurobehavioral tasks than Val/Val individuals during extended wakefulness. The neurotrophic BDNF is an established mediator of long-term synaptic plasticity, neurogenesis, and learning and memory [32].

Thus, the deleterious effects of TSD on neurobehavioral performances would not be based on a single genetic polymorphism, and few studies have evaluated the influence of multiple genetic polymorphisms when compared with vulnerability categories based on tertiles of change in cognitive performances during TSD.

Among available techniques to assess genetic polymorphism, the sequencing and the conventional TaqMan PCR methods are highly reliable [33]. There is also the Sequenom MassARRAY iPLEX Platform, based on distinguishing allele-specific primer extension products by mass spectrometry (MALDI-TOF), but conditioned by the molecular weights of alternative extension products. Finally, our lab has evaluated the LAMP-MC (loop-mediated isothermal amplification and melting curve analysis) method on blood and buccal cells, without prior DNA extraction, for detection of small SNPs sample size (*n* = 5) [34]. Another technique is the array-based hybridization, which consists of chips where specific probes of a panel of genes are attached.

The aim of our study was to compare influence of 14 SNPs on the performance degradation of sustained attention using PVT, executive function using the inhibition Go/noGo task, and subjective sleepiness in 47 healthy subjects during TSD. The SNPs selection was based on the studies cited above. The second objective was to evaluate influences of genetic polymorphisms that are immediately involved in vulnerability to sleep loss, as well as to distinguish between high vulnerability and high resistance profiles.

## 2. Materials and Methods

### 2.1. Subjects

Forty-seven healthy subjects (37.0 ± 1.1 years) were included in this study after providing their informed written consent. The study received the agreement of the Cochin–CPP Ile de France IV (Paris) Ethics Committee and of the French National Agency for Medicines and Health Products Safety (ANSM) (Ile de France IV) and was conducted according to the principles expressed in the Declaration of Helsinki of 1975, as revised in 2001. The study has been recorded in the clinical trial base (clinical trials.gov ID: NCT03859882).

Subjects were free from medical, psychiatric, and sleep disorders. Subjects had not travelled between time zones within 7 days prior the study. Exclusion criteria included physical or mental health troubles according to (I) Hospital Anxiety and Depression scale, HAD ≥16 [35]; (II) significant medical history; (III) Epworth Sleepiness Scale, ESS >11 [36]; (IV) Pittsburg sleep quality index, PSQI >8 [37]; (V) morningness–eveningness questionnaire <31 or >69 [38]; and (VI) habitual time in bed per night <6 h. Participants were excluded if they self-reported any use of medications with sleep-related side effects and illicit drugs. They completed a sleep/wake schedule for the week prior to the study.

### 2.2. Study Design and Testing Conditions

The 3-day in-laboratory experimental protocol included (I) a habituation/training day (D0), (II) a baseline day (D1), (III) a total sleep deprivation (TSD) day beginning on D1 at 7:00 until D2 at 21:00 (meaning 38 h of continuous wakefulness), and (IV) a recovery night until the end of the study (09:00 on D3).

Subjects were in individual temperature-controlled (22 ± 1 °C), 3 × 4 m rooms that included a bed, restroom facilities, and a computer workstation. Laboratory illumination was maintained at 150–200 lux during the entire period of sleep deprivation (with lights off during sleep periods). Subjects were asked to maintain regular caffeine consumption for the week prior to the experiment, and exercise, tobacco, and alcohol were prohibited 48 h before and during the study. Meals and caloric intake were standardized for all subjects (2600 kcal/day). When not engaged in any specific testing or meals, subjects followed a standardized activity program (reading, watching videos, and playing video games). At least 2 investigators were systematically present in the laboratory. Two teams of 12 h shifts were organized to maintain a good level of investigator alertness. When the subjects were about to fall asleep (eyes closed, head down), they were gently and immediately woken up (i.e., no period of sleep >30 s).

### 2.3. Procedures

The sleepiness scale (KSS), psychomotor vigilance test (PVT), and executive Go/noGo task were performed in this order at six times between 07:00 on baseline (D0) to 21:00 on D2 (i.e., after 2, 8, 14 h during D1, and 20, 26, 32 h of wakefulness during D2). All subjects had a systematic habituation period for tests at D0 (habituation/training day) in order to reduce a learning bias. During all the periods in the sleep laboratory, including the tests, actimetry activity was recorded to confirm that all subject stayed awake during the 38-h continuous wakefulness period.

### 2.4. Neurobehavioral Testing

#### 2.4.1. Karolinska Sleeping Scale (KSS) for Sleepiness

The KSS is a subjective scale used to grade the subject awakening from 1 to 9; 1 represents “extremely alert” and 9 represents “extremely sleepy” [39]. The computerized version used along this study enables the subject to choose out of the nine given options.

#### 2.4.2. Psychomotor Vigilance Task (PVT) for Sustained Attention

We utilized a computer-based version of the 10 min PVT. This test is easily reproducible, the number of trials per session reduces the hazard bias, and the data are simple enough to be processed efficiently. Subjects were asked to respond, by clicking the left mouse button, to the appearance of a visual stimulus (a millisecond counter) as quickly as possible without making false starts. The inter-stimulus interval, defined as the period between the last response and the appearance of the next stimulus, varied randomly from 2 to 10 s. The number of PVT lapses of attention is defined as reaction time >500 ms. The reaction time (RT) in milliseconds for a 1 s period and PVT response was regarded valid if RT was ≥100 ms. Results are expressed as the number of lapses and speed (1/reaction time × 1000) [6].

#### 2.4.3. The Go/NoGo Executive Task for Inhibition and Impulsivity

In this test, subjects were required to respond or not to a stimulus on a screen. After the appearance of a fixation cross in the center of the screen for 500 ms, an arrow appeared in the center of the screen for 1 s. Depending on the test instruction, subjects have 2 s to respond when the arrow pointed out on the right (“Go” response) and not to respond when it pointed on the left (“noGo“ response). The proportion is always as followed: 67% of “Go” trials and 33% of “noGo” trials. Subjects have 2 s to respond, and their response was directly followed by a new trial in order to determine the capacity to consciously inhibit non-relevant automated responses (inhibition process). The total duration of the task is around 7 min 30 s [40]. Performance of the task was assessed by calculating number of commission errors (ratio).

### 2.5. Saliva DNA Extract and Genotyping

Oragene DNA kits OG-500 (DNAgenotek, Kanata, ON, Canada) were used to collect whole saliva samples from healthy adult volunteers (*n* = 47) after rinsing the mouth with water and at least 30 min after eating or drinking. DNA from saliva collected in Oragene containers should be stable for at least 5 years at ambient temperature. After manual cell lysate preparation, the genomic DNA purification was performed by the Autopure LS instrument (Qiagen, Hilden, Germany). DNA quantity, integrity, and ability to PCR were evaluated by Quality Controls. Participants were genotyped by predesigned or customized probes, and TaqMan SNP genotyping assays were provided by Thermo Fisher Scientific (Whaltham, MA, USA). PCR was performed on GeneAmp PCR System 9700, and a 7900HT system with SDS software version 2.4 (Applied Biosystems, Foster City, CA, USA) was used for fluorescence detection and allelic discrimination. A check by Sanger sequencing of PCR products was carried out for rs5751876 because ADORA2A gene includes many SNPs in this region.

### 2.6. Statistical Analysis

Statistical analyses were computed using R-studio (V 0.99.902—2009–2016 RStudio, Inc., Boston, MA, USA). Values were expressed as mean ± 95% confidence interval (95% CI).

Effects of sleep deprivation on PVT, Go/noGo, and KSS score were assessed using a one-way repeated measure analyses of variance (ANOVA) followed by a post hoc comparison with baseline values at 2 h of wakefulness. The associations between SNPs were checked using a chi-squared test.

The primary objective was to demonstrate the differences between genetic polymorphisms on the cognitive parameters and KSS score at the nadir time point (at 8:30 on D2). Results were analyzed using an ANOVA, including fixed effects for genetic polymorphism (non-repeated measure) and time awake (repeated measure). If a genotype group of homozygous mutation counts less than 6 members, heterozygous and homozygous alleles were pooled for graphic and statistical analysis.

A one-way analysis of variance (ANOVA) was used to determine the percentage of variance that was explained by genotype, as well as to estimate Cohen’s local effect size f2, at the nadir of performance (at 8:30 on D2). Effect sizes of 0.0099, 0.0588, and 0.1379 were considered small, moderate, and large, respectively [41,42].

The subject-specific averages for vulnerability to sleep loss were analyzed using nonparametric one-way analysis of rank scores with genotype as independent variable. For reference, the sample was also divided into tertiles on the basis of subjects’ rank order of vulnerability (as previously quantified by averaging the results (PVT lapses number, KSS score and Go/noGo commission errors) over the 24 h period of sleep deprivation). The temporal profiles of the genotypes as revealed by the ANOVA were visually compared to the group-average temporal profiles of each of the tertiles [20].

For the 4 SNPs identified as having the largest effect on both performance tasks and KSS score during prolonged wakefulness, we conducted a combination analysis using SNPstats R software [43] to assess a potential synergistic combination of the mutations on the values observed at 26 h of prolonged wakefulness. The mean results for each allele combination, the mean difference, and the *p*-value of the comparison with the most frequent combination were calculated using a Bonferroni test.

## 3. Results

Subjects’ characteristics are presented in Table 1.

### 3.1. Genotypes Frequencies

The genotype prevalence of the 14 SNPs was most of the time similar to the 1000 Genomes Project data on the GRCh38 reference assembly (http://www.internationalgenome.org, last access the 10 October 2021) (Table 2), although some particularly rare mutations were not represented by a satisfactory number of subjects (≤6 people). Therefore, if group of homozygous mutation counted less than six members, heterozygous and homozygous alleles were grouped. In those cases, the legends provide the information. Genotype frequencies of all SNPs conformed to Hardy–Weinberg equilibrium (*p* > 0.16 for all). The pairwise linkage disequilibrium analysis showed moderate to strong LD between ADORA2A variants [12].

The IL-1β genotype (and five others, ADORA2A (rs2298383 and rs5751876), ADA, DRD2, and BDNF) of one individual were missing because undetected at the probe level, although the DNA was of good quality. This is inherent to the Taqman reference method, with 1 to 4%, depending on the polymorphisms, not detected.

### 3.2. SNPs Distribution Associations

Two SNPs of ADORA2A (rs5751862, rs2298383) were significantly associated with TNF-α (*p* = 0.004 and *p* = 0.030), one SNP (rs5751862) with IL1-β (*p* = 0.044), and four SNPs (rs5751862, rs2298383, rs4822492, and rs5751876) with COMT (*p* < 0.001 for the first three and *p* < 0.01 for the last). In addition, the IL1-β SNP was associated with ADA and COMT (*p* < 0.001 and *p* < 0.01) and IL-6 with BDNF (*p* < 0.01).

### 3.3. Correlations between Tests and Questionnaire Results

Pearson correlations analyses showed significant correlations with an R^2^ > 0.4 between PVT speed and PVT lapses (R^2^ = −0.42, *p* < 0.001). PVT lapses and PVT speed were correlated with KSS scores (R^2^ = −0.2, *p* ≤ 0.01 and R^2^ = −0.2, *p* < 0.01) and Go/noGo commission errors (R^2^ = −0.3, *p* < 0.01 and R^2^ = −0.2, *p* = 0.01). No significant correlation was observed between Go/noGo commission errors and KSS score (R^2^ = −0.02, *p* = 0.2).

### 3.4. Results of Awakening Effect on PVT, Go/NoGo, and KSS Parameters

The ANOVA statistical analysis evidenced a significant awakening effect with the decrease of sustained attention in the PVT illustrated by speed (F_46,6_ = 72.2, *p* < 0.001) and number of lapses (F_46,6_ = 16.8, *p* < 0.001), the decrease of inhibitory capacity in the Go/noGo task illustrated by commission errors (F_46,6_ = 9.3, *p* < 0.001), and the decrease of daytime sleepiness illustrated by the KSS score (F_46,6_ = 31.6, *p* < 0.001) (Figure 1).

The largest effect of prolonged wakefulness was observed after 26 h of prolonged wakefulness in comparison to 2 h of wakefulness for PVT lapses (mean difference = 7.4 ± 1.4, *p* < 0.001), PVT speed (−0.62 ± 0.09 s^−1^ × 1000, *p* < 0.001), and Go/noGo commission errors (4.5 ± 0.9 %, *p* < 0.01). For KSS, the greatest difference was observed after 20 h of prolonged wakefulness (−1.7 ± 0.2, *p* < 0.01). An awakening effect after 26 h of prolonged wakefulness was significant (*p* ≤ 0.001) for all 14 studied polymorphisms.

### 3.5. Results of Effects Size (ES) Analysis at Nadir Time Point (08:30) on Day 2 (i.e., 26 h of Continuous Awaking)

Figure 2 shows the large difference between mutated and ancestral alleles for the 14 SNPs on cognitive tasks and KSS score. The ES shows a large significant effect on PVT lapses for the rs5751862 of ADORA2A and TNF-α polymorphisms, as well as a medium significant effect for IL1-β and the rs2236624 of ADORA2A. We found that 16.1% of the total variance was explained by the rs5751862 of ADORA2A, 11.7% by TNF-α polymorphism, and 7.1% by IL1-β and 6.5% by rs2236624 of ADORA2A.

For the PVT speed, the ES showed a large significant effect for TNF-α and the rs3761422 of ADORA2A polymorphisms, and a medium significant effect for COMT and the rs5751862 of ADORA2A. We found that 10.3% of the total variance was explained by TNF-α polymorphism, 9.2% by the s3761422 of ADORA2A, 9.1% by COMT, and 7.8% by rs5751862 of ADORA2A.

For the Go/noGo commission errors, the ES showed a large significant effect for COMT; the rs5751862 of ADORA2A; and the three proinflammatory cytokines: IL-6, IL1-β, and TNF-α. We found that 14.4% of the total variance was explained by COMT polymorphism, 12.2% by the rs5751862 of ADORA2A, 10.1% by IL-6, and 9.1% by IL1-β.

For the KSS score, the ES showed a large significant effect for the ADA and the rs5751876 of ADORA2A, and a medium significant effect for TNF-α and IL-6 polymorphisms. We found that 14.4% of the total variance was explained by ADA polymorphism, 14.1% by rs5751876 of ADORA2A, 8.4% by TNF-α, and 8.3% by IL-6.

### 3.6. Results of Polymorphism Effect and Interaction with Awakening (SNP × TSD) on PVT, Go/noGo, and KSS Parameters

For the PVT, Go/noGo, and KSS parameters at the nadir time point, the degrees of freedom for the sleep deprivation effect and for polymorphism effect (SNP effect) were, respectively, F(5,42) and “F(2,46) or F(1,46)”. For the interaction (SNP × TSD), the degrees of freedom were F(10,46) or F(5,46).

*PVT performance*: On the PVT lapse number at the nadir time point, there was a significant SNP effect for 7 SNPs out of 14, including the three proinflammatory cytokines (TNF-α, IL1-β, and IL-6) and four SNPs of ADORA2A (rs5751862, rs4822492, rs2236624, and rs2298383) (Table 3). The SNP × TSD interaction was significant for rs5751862 and rs2236624 of ADORA2A and TNF-α (Table 3). Figure 3 represents the awakening response of the first four SNPs with large or medium ES, for the ancestral and mutated genotypes (if the mutated homozygous or heterozygous alleles were extremely rare for some polymorphisms (TNF-α and rs2236624 ADORA2A), mutated homozygous and heterozygous alleles were combined), and compared with the vulnerability categories based on tertiles of change, vulnerable, intermediate, and resilient. On the PVT speed, there is a significant SNP effect without interaction for TNF-α and IL-6, and significant interaction without SNP effect for DRD2 and BDNF (Table 3).

*Go/noGo performance*: At the nadir time point, there is a significant SNP effect for six SNPs (TNF-α and IL-6, ADORA2A (rs5751876 and rs2236624), ADA, and PER3) without significant interaction with TSD. A significant interaction without SNP effect was present for COMT and there was an SNP effect and interaction for PER3 (Table 3). Figure 4 represents the awakening response of the first four SNPs with large ES, COMT, rs5751862 ADORA2A, IL-6, and IL1-β.

*KSS score*: At the nadir time point, there was a significant SNP effect and interaction with TSD for TNF-α (Table 3). SNP effects without interaction were present for IL-6, the rs5751876 of ADORA2A, ADA, and COMT. Figure 5 represents the awakening response of the first four SNPs with large or medium ES, ADA, rs5751876 ADORA2A, TNF-α, and IL-6.

### 3.7. Results of Polymorphism Combination Analysis on PVT, Go/NoGo, and KSS Parameters

Table 4 and Figure 6 show that carriers of the allele combination GGGT (respectively, for TNF-α, IL1-β, and ADORA2A rs5751862, and T carriers of ADORA2A rs2236624) were significantly less degraded for PVT lapses compared to carriers of the most frequent allele combination AGAC (*p* = 0.02). This latter combination included A allele carriers of ADORA2A rs5751862 (representing ≈75% of our population) and IL1-β (≈83% of our population).

For Go/noGo commission errors, there were four allele combinations (GAGG, AAAA, ATGG, AAAG for IL1-β, IL-6, COMT, and ADORA2A rs5751862, respectively) that were significantly less degraded than the most frequent allele combination AAGG (Table 4, Figure 6). This latter combination included G allele carriers of COMT (representing ≈64% of our population) and A allele carriers of IL1-β (≈83% of our population). Among the four less degraded combinations, the AAAG combination included the A allele of COMT, A of IL-6, and G of ADORA2A rs5751862, and the AAAA combination included the A allele of COMT and A carriers of IL-6.

For KSS scores, there was one allele combination (GTCT—for G allele carriers of TNF-α, T carriers of IL-6, C carriers of ADA, and T carriers of ADORA2A rs5751876, respectively) that was significantly less degraded than the most frequent allele combination GACC (*p* = 0.03) (Table 4, Figure 6). This latter combination included C allele carriers of ADA (representing ≈93% of our population) and C allele carriers of ADORA2A rs5751876 (≈67% of our population).

## 4. Discussion

Our results indicate the neurobehavioral vulnerability/resilience of three genetic polymorphisms at the nadir of impairments corresponding to 26 h of continuous wakefulness, the rs5751862 of ADORA2A, TNF-α (rs1800629), and COMT (rs4680), in a group of 47 healthy subjects. Cognitive performance impairments and subjective sleepiness have been largely described during continuous wakefulness both in non-executive tasks such as psychomotor vigilance test (PVT) that specifically measures sustained attention processes and in the executive inhibition Go/noGo task [7,44]. We first illustrated in Figure 1 the individual variability in cognitive impairment and self-reported sleepiness assessed over consecutive 6 h intervals during the 38 h of continuous wakefulness with a significant increase in the number of PVT lapses and a decrease in speed, as well as an increase in commission errors in the Go/noGo inhibition executive task reflecting increased impulsivity to negative stimuli [45]. The effect size analysis between the cognitive and sleepiness impairments at the nadir and mutations on the 14 SNPs pointed large effects for PVT lapses with the rs5751862 of ADORA2A and TNF-α; for Go/noGo commission errors with COMT, the rs5751862 of ADORA2A and IL-6; and for KSS with ADA, the rs5751876 of ADORA2A and TNF-α. The comparison to the temporal profiles of the most vulnerable and resilient tertiles of the sample, regardless of genotype, revealed that carriers of the ancestral G/G genotype of rs5751862 ADORA2A was markedly close to the resilient profile for PVT lapses and that carriers of the ancestral G/G of rs5751862 SNP and ancestral G/G of COMT were, respectively, close to the most resilient and vulnerable for Go/noGo commission errors. Analysis of the four combined SNPs (i.e., those with the largest effect size on PVT lapses, IL1-β, TNF-α, ADORA2A rs5751862, and rs2236624) showed that the most resilient subjects were those carrying the GGGT combination including the G allele for ADORA2A rs5751862 and TNF-α. For Go/noGo commission errors, the combination of SNPs (IL1-β, IL-6, COMT, and ADORA2A rs5751862) closest to the resilient profile was AAAA, including the A allele for COMT and IL-6, and AAAG, including the A allele for COMT and IL-6 plus the G allele for ADORA2A rs5751862. For the KSS daytime sleepiness score, analysis combining TNF-α, IL-6, ADA, and rs5751876 ADORA2A showed that subjects carrying the G allele for TNF-α and the T allele for IL-6 (GTCT) were those that reported a significantly lower score compared to the most frequent and vulnerable genotypic profile.

Of the potential biological mechanisms underlying interindividual neurobehavioral vulnerability to sleep loss, the 14 SNPs selected here represented suitable candidates (see review by [9]), including adenosine-related genes (i.e., ADORA2A and ADA), dopamine-related genes (i.e., COMT and DRD2), TNF-α, and PER3. We previously evidenced the association between ADORA2A polymorphisms, particularly the most studied rs5751876, and the daily total sleep duration in a large group of 1023 active workers of European ancestry aged 18-60 years [15]. We also showed that the TNF-α polymorphism did not influence the TSD-related PVT lapse deficit, while the ADORA2A rs5751876 significantly interacted, but in a smaller group of 37 subjects, compared to the present study [19]. Thus, our results pertain to a large investigation on the potential implication of 14 SNPs on cognitive responses to total sleep deprivation, as well as the association with caffeine consumption.

The ADORA2A locus is located on human chromosome 22 and contains a set of SNPs, the most studied being rs5751876 (formally designated as 1976C/T or 1083C/T) and other polymorphisms such as rs5751862 and rs2236624, which are in high linkage disequilibrium with the rs5751876 [46]. These SNPs were identified and associated with the corresponding haplotypes with different anxiety-related personality scores [46,47,48,49]. The rs5751876 SNP is a synonymous variant located on the exon 4 position, while the rs5751862 is located on the 5′ promoter region; rs2298383 and rs3761422 are on 5′UTR, rs2236624 is intronic on 3-4, and rs4822492 is on 3′UTR. The 5′ promoter region usage can have a major impact on gene expression and can alter expression of the associated gene at both the mRNA and protein level in humans [50]. With respect to the A2A receptor, we previously showed that one night of TSD induced the upregulation of its gene expression in leukocytes from healthy subjects [51]. In our study, the rs5751862 is the primary SNPs of ADORA2A with large effect size with responses in the two cognitive tasks, and the ANOVA analysis in particular showed this SNP significant effect and significant interaction with TSD at the nadir of PVT lapse performance. At the nadir time point, carriers of the ancestral G/G genotype of ADORA2A rs5751862 were less degraded compared both to mutated G/A and A/A genotypes. Moreover, for PVT lapses, a significant effect of SNP was also observed for ADORA2A rs2236624 with a significant interaction with TSD, while the effect size was medium, and comparison with the time profiles of the most vulnerable and resilient tertiles on PVT lapses response revealed a modest influence of this SNP (i.e., subjects carrying the ancestral and mutated genotypes lying on either side of the intermediate subject response). Our results can be compared, with some reservations, to those of Bodenman et al. [17], showing that carriers of the HT4 haplotype of ADORA2A (present in 14 of 45 subjects), including G allele carriers of rs5751862, performed faster in PVT during TSD compared to carriers of non-HT4 haplotype. In this study, the best performing HT4 haplotype on PVT speed included C allele carriers of the rs2236624 ADORA2A, which in our study were in particular more degraded than T allele carriers on the lapses. The ANOVA analysis in our study did not show a significant SNP effect or interaction with TSD on any of the ADORA2A variants with respect to PVT speed. The difference in the number of subjects between our study and Bodenman’s study [17] may explain the difference in PVT speed response in carriers of the rs2236624 variant.

In our study, there were significant associations between the ADORA2A rs5751862 and the two pro-inflammatory cytokines, TNF-α (*p* = 0.004) and IL1-β (*p* = 0.044). The pro-inflammatory cytokine TNF-α is involved in sleep/wake regulation [21]. In theory, prolonged wakefulness leads to release of adenosine triphosphate (ATP) into the cellular space, which binds to the purine type 2 receptor, leading to a release of TNF-α and IL1-β, which can explain the crucial role in the regulation of the sleep/wake cycle of the pro-inflammatory cytokines and the adenosine receptors (ADORA2A). In our previous studies, we have described increases of TNF-α plasma levels and of A2A receptor expression in leukocytes after 36 h of continuous wakefulness in healthy men concomitantly with PVT lapse impairment [7,51,52]. On the basis of these results, we hypothesized that TNF-α polymorphism might influence the neurobehavioral response to TSD, as this polymorphism leads to increased TNF-α gene transcription [53] and TNF-α cytokine production [23]. In this study, the effect size analysis showed that TNF-α is positioned as the second SNP presenting a large effect size in PVT lapses, and IL1-β is the third SNP but with a medium effect size. Although TNF-α accounts for only a modest portion of the observed interindividual differences in temporal patterns of vulnerability/resilience in PVT (as described above for ADORA2A rs2236624), results of the effect-size analysis and the SNP x TSD ANOVA analysis clearly evidenced its influence along with the one of rs5751862 of ADORA2A. In this sample of 47 subjects, carriers of the ancestral G/G genotype of TNF-α were less degraded compared both to mutated G/A and A/A genotypes at the nadir of PVT lapse performance (at 08:30, corresponding to 26 h of continuous wakefulness) [6,54]. We also showed that the ancestral G/G genotypes are significantly less degraded than A allele carriers on the self-reported KSS score of sleepiness. The effect size was between medium and large, and a significant polymorphism effect and interaction with TSD was present in the ANOVA analysis. In the herein described results, we evidenced significant effect of TNF-α polymorphism on PVT lapses and sleepiness score that was not statistically present in our previous study involving 37 subjects [19].

In our study, there was additionally a strong significant association between the rs5751862 ADORA2A and COMT (*p* = 0.001), which is illustrated at the 26 h nadir of the executive Go/noGo inhibition task vulnerability/resilience temporal profiles, with homozygous mutated A/A of ADORA2A rs5751862 being highly sensitive to TSD as is the ancestral G/G of COMT (rs4680). The catechol-O-methyltransferase (COMT) gene, the major enzyme degrading catecholamines, particularly dopamine, has also been implicated in vulnerability to sleep loss (see the review in [9]). The SNP involves a valine (Val) to methionine (Met) substitution at codon 158 (Val158Met); the Met and Val alleles differentially affect COMT’s enzymatic activity and thus influence dopamine levels in the prefrontal cortex. The Met (A allele) allele reduces the enzymatic activity of COMT three- to fourfold relative to the Val (ancestral G allele), leading to increased dopamine availability. The COMT polymorphism has been shown to markedly affect enzyme activity, protein abundance, and protein stability [55,56]. In our study, with respect to PVT lapses and Go/noGo cognitive tasks, the SNP COMT is in a large effect size with the Go/noGo performance only, and the ANOVA analysis showed no significant SNP effect but significant interaction between SNP and TSD in the two cognitive tasks, indicating that the COMT genotype effect was specific to TSD. We previously evidenced in 37 subjects that A allele carriers of COMT are significantly more degraded than homozygous G/G ancestral carriers in the PVT task only after 20 h of continuous awaking, while no SNP difference occurred after 26 h [19]. In comparison, we showed here in 47 subjects that the A allele carriers were less vulnerable at 26 h of continuous wakefulness (corresponding to the nadir of performance) in the executive Go/noGo inhibition task. Using the Go/noGo reversal learning task, Satterfield et al. [25] previously demonstrated the considerable vulnerability in adaptive decision making for subjects with the Val (G allele) of COMT Val158Met, especially in subjects that were G/G homozygous. Our results confirmed the vulnerability for subjects carrying the G/G genotype of COMT in the executive inhibition task. The difference in the COMT genotype distribution between our two studies (29.7% carrying the G/G genotype in Erblang et al. [16] and 36.2% in this study) may explain why G/G was not more degraded than G/A and A/A at the nadir of performance regarding PVT lapse in Erblang et al. [19]. With respect to the dopaminergic system, the interaction of the D2 receptor (DRD2) polymorphism and the DAT1 transporter has been shown to be involved in modulating the neurobehavioral, subjective, and neurophysiological consequences of TSD [26]. In our study, the effect size was low for DRD2 (rs1079597) polymorphism with respect to parameters of cognitive responses and sleepiness score, and there was a significant interaction between the DRD2 SNP and TSD only for PVT speed.

In this study, the effect size and ANOVA analyses on the influence of the 14 SNPs on the neurobehavioral vulnerability to TSD evidenced the possible influence of the two other pro-inflammatory cytokines, IL1-β and IL-6, notably, the IL-6 SNP significant effect on PVT lapses and speed, Go/noGo commission errors, and KSS score. We showed that subjects carrying T allele of the IL-6 SNP (rs4719714) were more degraded than those carrying the ancestral A/A genotype at the nadir of Go/noGo inhibition task, at which time mutated ADORA2A rs5751862 subjects and the low activity dopamine-related G/G COMT ones were the most degraded. The IL-6 and IL1-β cytokines have some sleep regulation properties, and sleep deprivation increases daytime IL-6 and causes somnolence and fatigue during the next day [21,57]. The rs4719714 of IL-6 has been found in high LD with the -6331T>C (rs10499563), which is associated with increased IL-6 serum concentrations in an acute inflammatory state [58].

Among the 14 SNPs, the effect size are medium between PER3 and cognitive responses on psychomotor vigilance and executive inhibition tasks. However, the ANOVA analysis showed a PER3 SNP significant effect and interaction with TSD for the Go/noGo inhibition task, and a significant interaction with TSD without SNP effect on PVT lapses. PER3 is a circadian gene linked to diurnal preference. We confirmed a previous study showing that PER3 influences PVT lapses performance and sleepiness during TSD, comparing *n* = 14 carriers of the homozygous long allele (PER3^5/5^) and *n* = 15 carriers of the homozygous short allele (PER3^4/4^) [28]. In our previous study, no significant PER3 main effect nor interaction with TSD was observed on PVT lapses and subjective sleepiness on a group of 37 subjects [19]. As the proportion of carriers of the PER3 G allele was clearly lower than that of carriers of the C/C genotype (19.1% vs. 80.9%), it seems essential to increase the number of subjects before concluding on the influence of PER3 on cognitive performance related either to the sustained attention of the PVT or to the executive process of inhibition of Go/noGo.

Our results evidenced low effect size for the BDNF polymorphism with cognitive responses and sleepiness, with only one significant ANOVA interaction regarding PVT speed. The association analysis highlights its positive relationship with the IL-6 polymorphism (*p* < 0.01), which would merit further study in a larger number of subjects, especially since both polymorphisms are associated with the fatigue symptom [22].

Finally, our results showed the large effect size between ADA polymorphism and KSS sleepiness score, and its ANOVA main effect without interaction with TSD. ADA is an enzyme involved in the metabolism of adenosine that is thought to be implicated in homeostatic sleep/wake regulation. The T minor allele is associated with 35% lower levels of ADA, and carriers of the T allele show a higher level of both circulating adenosine and adenosine inside cells. This allele was shown to affect the duration and intensity of deep sleep and to reduce vigilant attention in humans [16,59]. The allele frequency distribution in our sample is in accordance with the expected data from the 1000 Genomes database. We illustrated the influence of the ADA polymorphism on the temporal vulnerability profile of the TSD sleepiness score despite the low percentage of the mutated genotype (6.5%) to show that carriers of the rare T allele are below the resilient profile, and thus should be discarded in future studies, especially those including small numbers of subjects, on the genetic determinants of neurobehavioral vulnerability to sleep loss.

It should be noted, as a limitation of our work, that the number of subjects remain limited. In any case, we must remain attentive to the multiple co-factors that modulate the neurobehavioral vulnerability to sleep loss and are sometimes more influent than the genetic impact, such as physical activity >2 h/week or coffee/energy drink consumption [60]. Future studies could try to recruit a larger population and more diversified profile population (age, gender, lifestyle habits, caffeine consumption). It would also be interesting in the future to focus on the influence of certain mutations on the basal attention level of the subject, without taking into consideration the sleep deprivation factor.

## 5. Conclusions

In conclusion, in this study, we showed that genetic polymorphisms in ADORA2A (rs5751862), TNF-α (rs1800629), and COMT are involved in creating profiles of high vulnerability or high resilience to sleep deprivation. These results could be used for the benefit of the Armed Forces to individualize and optimize countermeasure strategies to limit performance impairments induced by sleep deprivation. In a previous study, Maire et al. [28] evidenced that manipulation of the sleep homeostatic state affects sustained attention and sleepiness differentially on the basis of PER3-dependent vulnerability. As an example, we recently showed that six nights of extended sleep (approximately 1.2 h/night) protects against vigilance impairments during TSD and improves their recovery speed [6], which would deserve to be studied under the influence of the three genetic determinants. Furthermore, our findings revealed SNPs (TNF-α, IL-6, COMT, BDNF) that are also associated with common symptoms (i.e., pain, sleep disturbance, fatigue, affective and cognitive symptoms) across chronic conditions [22], arguing for their inclusion in the clinical research.

## Figures and Tables

**Figure 1 life-11-01110-f001:**
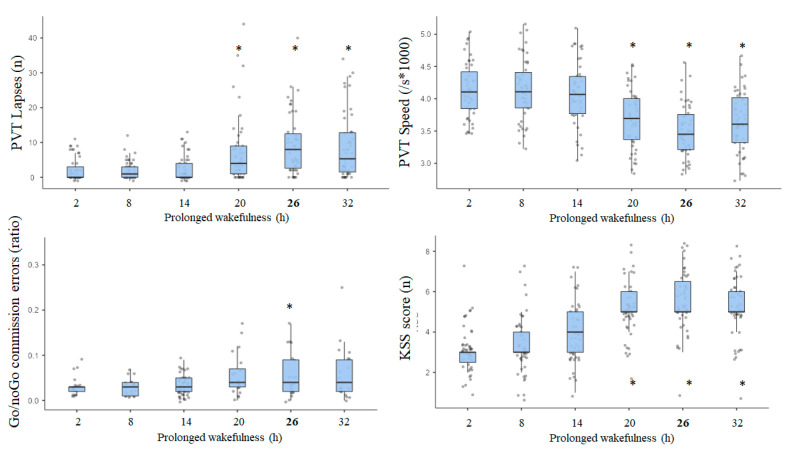
PVT lapses and speed, Go/noGo commission errors, and Karolinska sleepiness (KSS) scores during prolonged wakefulness. Results are median, 25% and 75% range, and individual points. * A significant difference with the values at 2 h of wakefulness.

**Figure 2 life-11-01110-f002:**
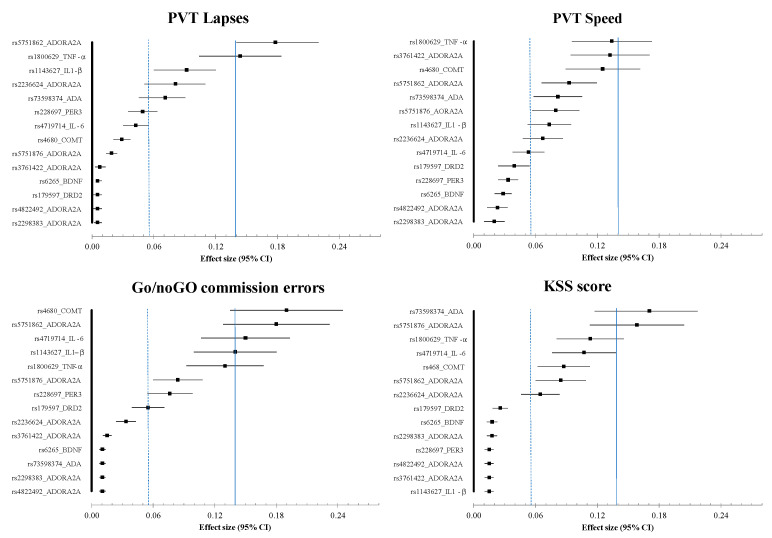
Effect size (absolute value) for genetic polymorphism mutations. Effect sizes of 0.0099, 0.0588, and 0.1379 were considered small, moderate, and large, respectively.

**Figure 3 life-11-01110-f003:**
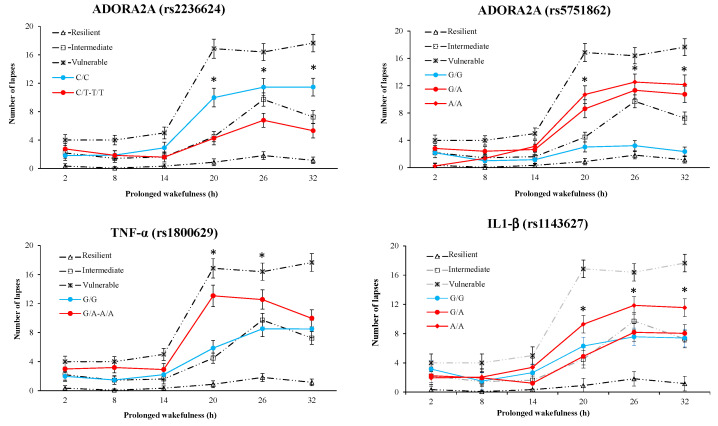
Average number of PVT lapses across consecutive 6 h intervals during 32 h of prolonged wakefulness for ADORA2A (rs2236624 and rs5751862), TNF-α, and IL1-β polymorphisms. For reference, the average number of lapses on PVT across consecutive 6 h intervals is shown for the most resilient, intermediate, and vulnerable tertiles of our sample without genotypes selection. * A significant difference with the ancestral profile (blue line).

**Figure 4 life-11-01110-f004:**
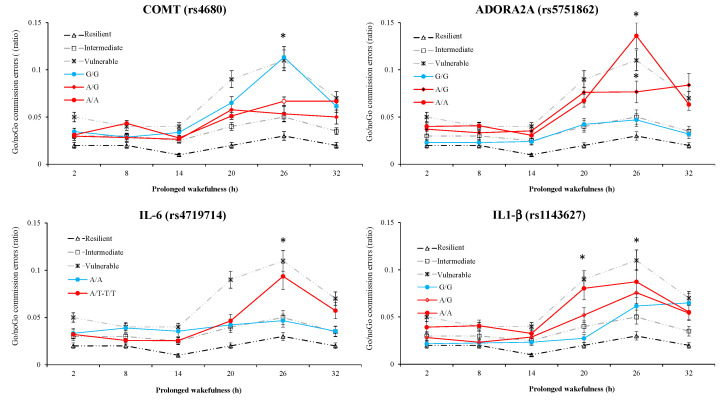
Average commission errors (ratio) on Go/noGo task across consecutive 6 h intervals during 32 h of prolonged wakefulness for COMT, ADORA2A (rs5751862), IL-6, and IL1-β polymorphisms. For reference, the average number of commission errors on Go/noGo across consecutive 6 h intervals is shown for the most resilient, intermediate, and vulnerable tertiles of our sample without genotypes selection. * A significant difference with the ancestral profile (blue line).

**Figure 5 life-11-01110-f005:**
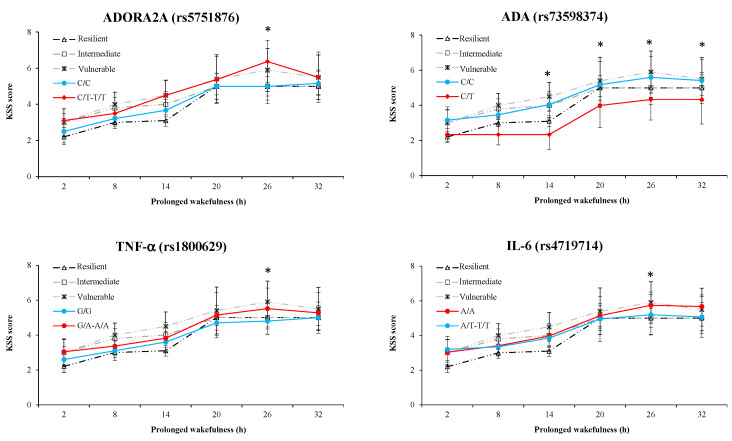
Average KSS (Karolinska Sleepiness Scale) scores across consecutive 6 h intervals during 32 h of prolonged wakefulness for ADORA2A (rs5751876), ADA, TNF-α, and IL-6 polymorphisms. For reference, the average KSS score across consecutive 6 h intervals is shown for the most resilient, intermediate, and vulnerable tertiles of our sample without genotype selection. * A significant difference with the ancestral profile (blue line).

**Figure 6 life-11-01110-f006:**
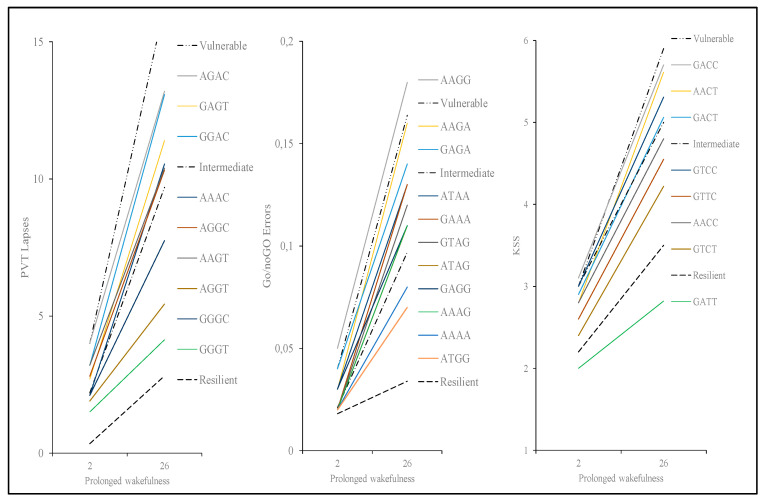
Average PVT lapses, Go/noGo commission errors, and KSS score after 2 h and 26 h of prolonged wakefulness for each SNP combination (see Table 4).

**Table 1 life-11-01110-t001:** Subjects’ characteristics.

Parameters	Values
Age (years)	37.0 ± 1.1
Weight (kg)	71.1 ± 1.4
Height (m)	1.7 ± 0.1
BMI (kg.m^−2^)	24.3 ± 0.7
Daily caffeine consumption (mg)	288 ± 17
Total sleep time (TST) (h) ^1^	7.1 ± 0.4
Physical exercise (h/week)	3.7 ± 0.3
Epworth sleepiness scale	7.8 ± 0.7

Values are mean ± 95% confidence interval (95% CI). ^1^ Mean TST value from the sleep/wake schedule completed the week before the study.

**Table 2 life-11-01110-t002:** Genetic distribution of polymorphisms in our population and in the data base of 1000 genomes.

Genetic Polymorphisms	Genotypes	Observed Count N (%)	Genotypes (Considered for Analysis) ^a^	N (%)	Expected Count from 1000 Genomes ^b^
					(%)
rs1800629_TNF-α	G/G (ancestral)	36 (76.6%)	G/G (ancestral)	36 (76.6%)	74.4%
(6:31.575.254)	G/A	10 (21.3%)	G/A–A/A	11 (23.4%)	24.5%
	A/A	1 (2.1%)			1.2%
rs1143627_IL1-β	G/G (ancestral)	8 (17.4%)			12.3%
(2:112.836.810)	G/A	18 (39.1%)			45.7%
	A/A	20 (43.5%)			41.9%
rs4719714_IL-6	A/A (ancestral)	27 (57.4%)	A/A (ancestral)	27 (57.4%)	59.8%
(7:22.721.094)	A/T	15 (31.9%)	A/T–T/T	20 (42.6%)	34.4%
	T/T	5 (10.6%)			5.8%
rs5751862_ADORA2A	G/G (ancestral)	12 (25.5%)			25.0%
(22:24.406.596)	G/A	27 (57.4%)			50.5%
	A/A	8 (17.1%)			24.5%
rs2298383_ADORA2A	C/C (ancestral)	7 (15.2%)			17.1%
(22:24.429.543)	C/T	25 (54.3%)			48.5%
	T/T	14 (30.5%)			34.4%
rs3761422_ADORA2A	C/C (ancestral)	16 (34.0%)	C/C (ancestral)	16 (24.0%)	38.4%
(22:24.430.704)	C/T	27 (57.5%)	C/T–T/T	31 (36.0%)	46.3%
	T/T	4 (8.5%)			15.3%
rs2236624_ADORA2A	C/C (ancestral)	27 (57.4%)	C/C (ancestral)	27 (57.4%)	54.7%
(22:24.440.056)	C/T	19 (40.4%)	C/T–T/T	20 (42.6%)	36.8%
	T/T	1 (2.2%)			8.5%
rs5751876_ADORA2A	C/C (ancestral)	15 (32.6%)	C/C (ancestral)	15 (32.6%)	37.4%
(22:24.441.33)	C/T	27 (58.7%)	C/T–T/T	31 (67.4%)	47.1%
	T/T	4 (8.7%)			15.5%
rs4822492_ADORA2A	C/C (ancestral)	7 (14.9%)			17.1%
(22:24.447.626)	C/G	26 (55.3%)			48.5%
	G/G	14 (29.8%)			34.4%
rs73598374_ADA	C/C (ancestral)	43 (93.5%)			88.1%
(20:44.651.586)	C/T	3 (6.5%)			11.5%
rs228697_PER3	C/C (ancestral)	38 (80.9%)	C/C (ancestral)	38 (80.9%)	81.7%
(1:7.827.519)	C/G	8 (17.0%)	C/G–G/G	9 (19.1%)	17.3%
	G/G	1 (2.1%)			1.0%
rs4680_COMT	G/G (ancestral)	17 (36.2%)			26.4%
(22:19.963.748)	G/A	18 (38.3%)			47.1%
	A/A	12 (25.5%)			26.4%
rs1079597_DRD2	C/C (ancestral)	32 (69.6%)	C/C (ancestral)	32 (69.6%)	73.4%
(11:113.425.564)	C/T	13 (28.2%)	C/T–T/T	14 (30.4%)	23.5%
	T/T	1 (2.2%)			3.2%
rs6265_BDNF	C/C (ancestral)	32 (69.6%)	C/C (ancestral)	32 (69.6%)	64.2%
(11:27.658.369)	C/T	13 (28.2%)	C/T–T/T	14 (30.4%)	32.2%
	T/T	1 (2.2%)			3.6%

^a^ Homozygous mutation (≤6 participants) was combined with heterozygous mutation. ^b^ Expected on the basis of 1000 Genomes Project data on the GRCh38 reference assembly (http://www.internationalgenome.org, accessed on 10 October 2021).

**Table 3 life-11-01110-t003:** ANOVA analysis of genetic polymorphism (SNP) effect and interaction with total sleep deprivation (TSD) on PVT (psychomotor vigilance task) and executive Go/noGo (inhibition task) parameters and KSS (Karolinska Sleepiness Score) at the nadir time point. In bold, significant results (*p* < 0.05).

	PVT_Lapses		PVT_Speed		Go/NoGo		KSS	
SNPs	SNP Effect	Interaction (SNP × TSD)	SNP Effect	Interaction (SNP × TSD)	SNP Effect	Interaction (SNP × TSD)	SNP Effect	Interaction (SNP × TSD)
rs1800629_TNF-α	*p* = 0.020	*p* = 0.030	*p* = 0.040	*p* = 0.250	*p* = 0.001	*p* = 0250	*p* = 0.010	*p* = 0.001
rs1143627_IL1-β	*p* = 0.030	*p* = 0.260	*p* = 0.400	*p* = 0.570	*p* = 0.600	*p* = 0.400	*p* = 0.080	*p* = 0.320
rs4719714_IL-6	*p* = 0.040	*p* = 0.930	*p* = 0.040	*p* = 0.570	*p* = 0.010	*p* = 0.280	*p* = 0.040	*p* = 0.470
rs5751876_ADORA2A	*p* = 0.200	*p* = 0.020	*p* = 0.390	*p* = 0.440	*p* = 0.001	*p* = 0.670	*p* = 0.010	*p* = 0.940
rs5751862_ADORA2A	*p* = 0.001	*p* = 0.018	*p* = 0.170	*p* = 0.210	*p* = 0.390	*p* = 0.090	*p* = 0.240	*p* = 0.430
rs4822492_ADORA2A	*p* = 0.040	*p* = 0.680	*p* = 0.500	*p* = 0.170	*p* = 0.510	*p* = 0.060	*p* = 0.690	*p* = 0.450
rs2236624_ADORA2A	*p* = 0.006	*p* = 0.007	*p* = 0.390	*p* = 0.240	*p* = 0.010	*p* = 0.060	*p* = 0.990	*p* = 0.450
rs3761422_ADORA2A	*p* = 0.430	*p* = 0.620	*p* = 0.390	*p* = 0.250	*p* = 0.790	*p* = 0.170	*p* = 0.640	*p* = 0.510
rs2298383_ADORA2A	*p* = 0.050	*p* = 0.680	*p* = 0.500	*p* = 0.810	*p* = 0.560	*p* = 0.060	*p* = 0.720	*p* = 0.400
rs73598374_ADA	*p* = 0.460	*p* = 0.790	*p* = 0.620	*p* = 0.770	*p* < 0.001	*p* = 0.520	*p* = 0.002	*p* = 0.610
rs4680_COMT	*p* = 0.110	*p* = 0.030	*p* = 0.350	*p* = 0.230	*p* = 0.350	*p* = 0.050	*p* = 0.020	*p* = 0.520
rs1079597_DRD2	*p* = 0.430	*p* = 0.760	*p* = 0.590	*p* = 0.040	*p* = 0.710	*p* = 0.520	*p* = 0.630	*p* = 0.220
rs6265_BDNF	*p* = 0.430	*p* = 0.520	*p* = 0.380	*p* = 0.040	*p* = 0.270	*p* = 0.710	*p* = 0.540	*p* = 0.210
rs228697_PER3	*p* = 0440	*p* = 0.001	*p* = 0.400	*p* = 0.810	*p* = 0.007	*p* = 0.030	*p* = 0.550	*p* = 0.880

**Table 4 life-11-01110-t004:** Analysis of SNP combination on PVT and Go/noGo tasks and KSS score. For each combination, the 4 SNPs were selected from the effect size analysis.

**PVT Lapses**								
	**rs1143627** **IL-1β**	**rs1800629** **TNF-α**	**rs5751862** **ADORA2A**	**rs2236624** **ADORA2A**	**Frequency (%)**	**Value** **(mean ± SD)**	**Difference** **(95% CI)**	***p*-Value**
1	A	G	A	C	25.3	12.2 ± 1.1		---
2	A	G	G	C	16.2	10.3 ± 0.5	−2.8 (−8.5–2.8)	0.32
3	G	G	G	C	15.2	7.8 ± 0.7	−5.4 (−1.7–0.8)	0.09
4	G	G	A	C	11.4	13.1 ± 0.8	−0.1 (−6.9–6.6)	0.97
5	A	G	G	T	9.6	5.4 ± 0.4	−7.7 (−14.0–−0.5)	0.04
6	G	G	G	T	9.1	4.1 ± 1.5	−9.1 (−15.2–−0.9)	0.02
7	A	A	A	C	9.1	14.9 ± 1.1	2.7 (−4.5–9.9)	0.48
8	A	A	G	T	2.3	10.4 ± 0.8	−2.8 (−5.2–9.6)	0.66
9	G	A	G	T	1.6	11.4 ± 0.8	1.8 (−5.82–19.4)	0.84
**Go/noGo Commission Errors**								
	**rs1143627** **IL-1β**	**rs4719714** **IL-6**	**rs4680** **COMT**	**rs5751862** **ADORA2A**	**Frequency** **(%)**	**Value (mean ± SD)**	**Difference** **(95% CI)**	***p*-Value**
1	A	A	G	G	14.3	0.18 ± 0.03		
2	G	A	G	G	13.0	0.11 ± 0.03	−0.08 (−0.13–−0.02)	<0.01
3	A	A	G	A	11.9	0.16 ± 0.02	−0.03 (−0.12–0.04)	0.49
4	A	A	A	A	10.2	0.08 ± 0.01	−0.10 (−0.17–−0.03)	<0.01
5	G	A	G	A	8.8	0.14 ± 0.02	−0.04 (−0.11–0.03)	0.29
6	A	T	A	A	8.3	0.13 ± 0.01	−0.05 (−0.11–0.01)	0.09
7	A	T	G	G	7.4	0.07 ± 0.01	−0.10 (−0.17–-0.03)	<0.01
8	G	A	A	A	6.6	0.13 ± 0.02	−0.05 (−0.13–0.02)	0.16
9	G	T	A	G	6.4	0.12 ± 0.03	−0.06 (−0.12–0.01)	0.11
10	A	A	A	G	6.3	0.11 ± 0.02	−0.08 (−0.15–−0.01)	0.03
11	A	T	A	G	4.5	0.11 ± 0.01	−0.07 (−0.16–0.01)	0.08
**KSS Score**								
	**rs1800629** **TNF-α**	**rs4719714** **IL-6**	**rs73598374** **ADA**	**rs5751876** **ADORA2A**	**Frequency** **(%)**	**Value** **(mean ± SD)**	**Difference** **(95% CI)**	***p*-Value**
1	G	A	C	C	50.6	5.7 ± 0.5	0	---
2	G	T	C	C	17.8	5.3 ± 0.4	−0.4 (−1.2–0.4)	0.36
3	G	A	C	T	8.8	5.1 ± 0.4	−0.6 (−1.9–0.6)	0.31
4	A	A	C	C	8.7	4.8 ± 0.3	−0.9 (−2.1–0.3)	0.16
5	G	T	C	T	7.5	4.2 ± 0.4	−1.5 (−2.8–-0.2)	0.03
6	A	A	C	T	4.1	5.8 ± 0.5	0.1 (−1.8–1.9)	0.92
7	G	A	T	T	1.3	2.8 ± 0.3	−2.9 (−6.2–0.4)	0.09
8	G	T	T	C	1.3	4.6 ± 0.3	−1.15 (−3.9–1.6)	0.42

## Data Availability

Data could be obtained by asking the corresponding author.

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
