# Peer review of "Genetics and Cognitive Vulnerability to Sleep Deprivation in Healthy Subjects: Interaction of ADORA2A, TNF-α and COMT Polymorphisms"

_life, 2021, doi:10.3390/life11101110_

Round 1

Reviewer 1 Report

In the present study, the authors investigated the role of various different single nucleotide polymorphisms (SNPs) on cognitive measures (a psychomotor vigilance task, a Go/noGo task and the Karolinska Sleepiness Scale) across a prolonged period of wakefulness for 32 hours. They found a general decrease in performance on these tasks over prolonged wakefulness and identified specific genetic polymorphisms that contribute to this effect. Specifically, the genetic polymorphisms in ADORA2A (rs5751862), TNF-α (rs1800629) and COMT appear to be involved in high vulnerability or high resilience to sleep deprivation. The study appears to be well designed and controlled and the topic investigated is of general interest. However, the novel contributions of this study are not entirely clear, and analyses can, at the time being, only be considered preliminary. Measuring multiple SNPs in single participants is a strength of the study, however, the authors do not perform any analysis investigating effects of combinations of the investigated SNPs. In addition, the manuscript suffers from some language issues that are, in parts, confusing. Please find below a list of major and minor issues:

Major issues:

- The aim of the study should be better defined in the abstract and introduction. The study appears to be mostly a replication of previous findings, especially for the vigilance and Go/noGo tasks. The authors should elaborate on their study’s contribution in advancing the existing literature.

- The discussion could be more concise and focused. What are the implications of these findings for the general population as well as for this research field?

- In the statistical analysis section, the authors state that “The primary objective was to demonstrate the differences between genetic polymorphisms on the cognitive parameters and sleepiness at the nadir time point (at 8:30 on D1)” (lines 223 ff.). It is unclear if the authors are in fact referring to 8:30 h on Day 1 or rather Day 2 (as they refer to in the remainder of their manuscript). Also, the rationale behind taking only one time point into consideration for these analyses is unclear and should be elaborated on. Why is this specific time point of special interest? It is also unclear which measures were taken in order to determine what time point is actually considered the “nadir time point”. How does this time point differ statistically from the other time points?

- In Table 2, some of the “observed counts” (3rd column) do not add up to 47 participants (e.g., the second genetic polymorphism [rs1143627_IL1-β] has 8 G/G, 18 G/A and 20 A/A genotypes, adding up to N = 46 instead of 47). Please clarify. Moreover, Table 2 suggests that several participants did not only have one but several SNPs. If this is, in fact, the case, it is likely that different SNPs might have additive effects on the cognitive measures. Can the authors differentiate effects of different SNPs on these measures?  It would also be interesting to investigate whether different SNPs have additive or synergistic effects on cognitive performance.

- It would be interesting to calculate how much of the variance in the cognitive and sleepiness responses to sleep deprivation can be explained by the single SNPs as well as by combinations thereof.

- Participants were sleep deprived until 21:00 h of Day 3. However, cognitive measures were only taken until 6 hours earlier (15:00 h). The authors should clarify why they did not collect data until the end of sleep deprivation. Given that participants spent another night in the laboratory following sleep deprivation (i.e., a “recovery night”), it would be very interesting to see if participants’ performance on the cognitive tasks renormalized and if this differed between different genetic polymorphisms. Is it possible to provide such data?

- The authors appear to have calculated mixed-effects models which differ from an analysis of variance (ANOVA). However, they use these terms interchangeably (e.g., lines 225 ff.), leading to confusion about what statistical modeling they did use. They should clarify this in the Methods section.

- In the Results section 3.4., the outcome of the statistical analysis is unclear and only one p value is given suggesting an “awakening effect” after 26 hours of prolonged wakefulness. It is unclear what measures (PVT, Go/noGo, KSS) this test refers to and why the authors only focused on this one time point in their analyses. They should provide a more detailed analysis in this section, including main and interaction effects for time awake and performance in the different tasks. Did the authors calculate whether the results of the cognitive tasks and sleepiness differed from baseline (i.e., before sleep deprivation)?

- The authors used Cohen’s d as a measure of effect sizes to be reported. However, the cutoffs used for the effect sizes (Cohen’s d > 0.01 = small, > 0.056 = medium, and > 0.14 = large) do not match with the recommended cutoffs of 0.2, 0.5 and 0.8 as small, medium and large effect sizes for this measure in the literature. The authors should clarify how they chose the effect size cutoffs for their study.

- Why did the authors use a cutoff at > 8 for the Pittsburgh Sleep Quality Index (PSQI), although it has been suggested by the developers of this questionnaire (Buysse et al., 1989) to set the cutoff already at > 5?

- The authors should elaborate on the difference between “predesigned” or “customized probes” participants were genotyped with (lines 213 f.). It is unclear why they used different probes for genotyping and how many participants were genotyped with one or the other. What is the difference between these probes? Are they equally well suited for this purpose?

- The authors collected actimetric data to confirm that participants stayed awake during the sleep deprivation protocol. They should indicate if this was in fact the case for all participants by providing some data. Similarly, participants were asked to fill a sleep/wake schedule for one week prior to the study. However, no results are provided concerning this measure. Did participants keep a regular sleep/wake cycle during this time?

- In the Results section “3.2. SNPs distribution associations”, it is unclear how the associations between SNPs with TNF-α, IL1-β, and COMT, as well as between IL1-β SNP and ADA and COMT, and TL-6 and BDNF were assessed. Please clarify.

- In the Results section “3.3. Correlations between tests and questionnaire results”, the authors should provide correlation coefficients and p values for each analysis individually rather than providing summarized statistics.

- Values for PVT speed are usually not normally distributed and strongly skewed. Yet, Pearson correlations were used for analyses. Please explain why using this correlation, instead of e.g., Spearman correlation, is appropriate for this data.

- In Figure 2, the authors provide effect sizes of genetic polymorphisms in the different tasks. However, the direction of these effects is unclear (i.e., whether the effect refers to a decrease or improvement in performance). It should also be added whether the effects were significant or not with respect to a specified reference.

- While caffeine intake was prohibited 48 hours before and during the study, participants’ regular caffeine consumption was rather high at 288 ± 17 mg per day. Given that the withdrawal effect is differently affecting individuals, usually kicking in 1-2 days after stopping caffeine consumption and lasts at least several days, this might have influenced the outcome of the study, which major purpose was to investigate cognitive measures that are known to be affected by caffeine and caffeine withdrawal, possibly interacting with the effects of SNPs. Please discuss.

- On Day 1 of the 3 days in the laboratory, total sleep deprivation began at 7:00 AM. Was this time matched with participants preferred wakeup time (e.g., based on the score in the morningness-eveningness questionnaire or the one-week sleep schedule)?

- It is unclear why only some SNPs are shown in Figures 3 – 5. In addition, the authors should provide detailed statistical analyses of the data shown in these figures. How do the SNPs of interest compare to each other and with respect to the tertiles?

- The language is sometimes difficult to understand. E.g., what do the authors mean with “constant causality relation” (e.g., line 282 or lines 313 f.) and “… target polymorphisms that are immediately involved in vulnerability to sleep loss” (lines 130 f.)?

- Line 242: What do the authors mean by “controlling for study”?

Minor issues

- The participants’ age in Table 1 (37.0 ± 1.1 years) does not match the number reported in the Abstract and Methods (38.2 ± 2.1 years). Please correct.

- The naming of the repeated-measures factor “awakening time” (e.g., line 227) is misleading because it implies that participants were woken up during these times. I suggest the authors to rename this factor to “time awake” or similar to avoid confusion.

- What do the authors mean by “approximately” after 2, 8, 14 hours etc. (line 171)? How much deviation was there between individual participants?

- I found it untypical to report degrees of freedom for the individual analyses in the Methods section. The authors should consider moving these measures to the respective results parts. It is also not clear why the degrees of freedom are different for the different measures. Please specify.

- What was the reasoning behind the authors’ decision to only include participants who scored between 31 and 69 in the morningness-eveningness questionnaire?

- There are several typos in the manuscript.

E.g., abstract: the acronym “SDT” should probably read “TSD” (total sleep deprivation)?

Abstract (line 64): should read “ADORA2A”.

Lines 200 f.: should read “Subjects have 2 s to respond…”.

Lines 217 f.: should read “… sequencing of PCR products was carried out for…”.

Line 232: should read “… and for polymorphism effect (SNP effect) are respectively …”.

Line 233: should read “F(2,46) or F(1,46).”.

Line 258: should read “Therefore, if group of homozygous mutations counts…”.

Title of Table 2 (line 277): should read “… in the data base of 1000 genomes”.

Author Response

In the present study, the authors investigated the role of various different single nucleotide polymorphisms (SNPs) on cognitive measures (a psychomotor vigilance task, a Go/noGo task and the Karolinska Sleepiness Scale) across a prolonged period of wakefulness for 32 hours. They found a general decrease in performance on these tasks over prolonged wakefulness and identified specific genetic polymorphisms that contribute to this effect. Specifically, the genetic polymorphisms in ADORA2A (rs5751862), TNF-α (rs1800629) and COMT appear to be involved in high vulnerability or high resilience to sleep deprivation. The study appears to be well designed and controlled and the topic investigated is of general interest. However, the novel contributions of this study are not entirely clear, and analyses can, at the time being, only be considered preliminary. Measuring multiple SNPs in single participants is a strength of the study, however, the authors do not perform any analysis investigating effects of combinations of the investigated SNPs. In addition, the manuscript suffers from some language issues that are, in parts, confusing. Please find below a list of major and minor issues:

We thank the reviewer and as recommended, to clarify the novelty of this study, we have added information in the introduction about the interest of targeting the effects of the presence of SNPs specifically involved in sleep/wake homeostasis and cognitive tolerance to total sleep deprivation (TSD) in single young and healthy participants. We also added the analysis of SNPs combination on levels of cognitive impairments and subjective sleepiness related to TSD.

Major issues:

- The aim of the study should be better defined in the abstract and introduction. The study appears to be mostly a replication of previous findings, especially for the vigilance and Go/noGo tasks. The authors should elaborate on their study’s contribution in advancing the existing literature.

We thank the reviewer. As recommended, we added information on the aim of the study in the abstract and the first paragraph of introduction. In the abstract we added “Common metrics of cognitive functioning for classifying vulnerable and resilient individuals in-clude the Psychomotor Vigilance Test (PVT), Go/noGo executive inhibition task and subjective daytime sleepiness.”

Our study is an extension on 47 subjects instead of 37 in our previous study (Erblang et al., 2021) for the vigilance PVT task only. The results on the Go/noGo task are original. The Go-noGo executive task evaluates capacity to withhold an automatic response, and increased commission errors for noGo stimuli traduced enhanced impulsivity to negative stimuli (Horn et al., 2003, Rabat et al., 2016). In addition, 14 SNPs instead of 4 were investigated including five additional SNPs for ADORA2A and the ones of the pro-inflammatory IL-1β and IL-6 cytokines. Our study advances the existing literature on the influences of genetic polymorphisms on cognitive vulnerability or resilience to TSD in healthy individuals by extending the influence of the pro-inflammatory IL-1β and IL-6 cytokines and strengthening those of ADORA2A, TNF-α and COMT.

  • Horn, N.R.; Dolan, M.; Elliot, R.; Deakin, J.F.W.; Woodruuf, P.R.W. Response inhibition and impulsivity: an FMRI study. Neuropsychologia 2003, 41, 1959–1966. doi: 10.1016/s0028-3932(03)00077-0.
  • Rabat, A.; Gomez-Merino, D.; Roca-Paixao, L.; Bougard, C.; Van Beers, P.; Dispersyn, G.; Guillard, M.; Bourrilhon, C.; Drogou, C.; Arnal, P.J.; Sauvet, F.; Leger, D.; Chennaoui, M. Differential Kinetics in Alteration and Recovery of Cognitive Processes from a Chronic Sleep Restriction in Young Healthy Men. Front. Behav. Neurosci. 2016;10:95. doi:10.3389/fnbeh.2016.00095

- The discussion could be more concise and focused. What are the implications of these findings for the general population as well as for this research field?

As recommended, we have made efforts to be more concise and reduce the discussion, particularly for the DRD2 and BDNF polymorphisms. On the other hand, we have brought information about the SNPs combination analysis in the Discussion. The implication of our findings is to add information about genetic predictors of cognitive vulnerability to sleep loss in a healthy young population, given that diminished neurobehavioral functioning may negatively impact productivity and performance in a variety of real-world settings, such as civilian populations or military operational personnel’s working under sleep debt. In addition, the consideration of potential genetic influences in the operational performance of sleep-deprived subjects would allow the military command to individualize advice in terms of countermeasures to deficits such as napping or banking sleep. This has been added in the conclusion as follow:

“These results could be used for the benefit of the Forces to individualize and optimize countermeasure strategies to limit performance impairments induced by sleep deprivation. In a previous study, Maire et al. [23] evidenced that manipulation of the sleep homeostatic state affects sustained attention and sleepiness differentially based on PER3-dependent vulnerability. As an example, we recently showed that six nights of extended sleep (approximately 1.2 h/night) protects against vigilance impairments during TSD and improves their recovery speed [6] (Arnal et al., 2015), which would deserve to be studied under the influence of the three genetic determinants. Furthermore, our findings revealed SNPs (TNF-α, IL-6, COMT, BDNF) that are also associated with common symptoms (i.e., pain, sleep disturbance, fatigue, affective and cognitive symptoms) across chronic conditions (Knisely et al., 2019), arguing for their inclusion in the clinical research”.

  • Knisely MR, Maserati M, Heinsberg LW, Shah LL, Li H, Zhu Y, Ma Y, Graves LY, Merriman JD, Conley YP. Symptom Science: Advocating for Inclusion of Functional Genetic Polymorphisms. Biol Res Nurs 2019 Jul;21(4):349-354. doi: 10.1177/1099800419846407. Epub 2019 Apr 25.

- In the statistical analysis section, the authors state that “The primary objective was to demonstrate the differences between genetic polymorphisms on the cognitive parameters and sleepiness at the nadir time point (at 8:30 on D1)” (lines 223 ff.). It is unclear if the authors are in fact referring to 8:30 h on Day 1 or rather Day 2 (as they refer to in the remainder of their manuscript). Also, the rationale behind taking only one time point into consideration for these analyses is unclear and should be elaborated on. Why is this specific time point of special interest? It is also unclear which measures were taken in order to determine what time point is actually considered the “nadir time point”. How does this time point differ statistically from the other time points?

We thank the reviewer and we apologize for the mistake: it is indeed “at 8:30 on D2”. Our previous results and others have evidenced that sleep deprivation-induced cognitive performance deficits on the PVT are typically most severe in the early morning (approximately 9:00 a.m.) due to continuous awakening (sleep pressure) and the nadir of the circadian rhythm of performance (Wesensten et al., 2002; Arnal et al., 2015; Sauvet et al., 2019).

Moreover, in order to improve our manuscript we show the results of the post-hoc analysis in the figure 1 and we describe the results in the dedicated paragraph. For PVT and Go/noGo results, we observed at 26 h of awaking the larger differences in comparison to values at 2 hours of awaking. This confirm the precedent results observed in our laboratory and literature. Moreover, we added in Results that the ANOVA statistical analysis showed a significant global awaking (TSD) effect for the 14 SNPs on PVT number of lapses, speed, Go/noGo commission errors and KSS score (p < 0.01 for all), and we added it in the Figure 1.

  • Wesensten NJ , Belenky G, Kautz MA, Thorne DR, Reichardt RM, Balkin TJ. Maintaining alertness and performance during sleep deprivation: modafinil versus caffeine Psychopharmacology (Berl). . 2002 Jan;159(3):238-47. doi: 10.1007/s002130100916. Epub 2001 Oct 19.
  • Arnal, P.J.; Sauvet, F.; Leger, D.; van Beers, P.; Bayon, V.; Bougard, C.; Rabat, A.; Millet, G.Y.; Chennaoui, M. Benefits of Sleep Extension on Sustained Attention and Sleep Pressure Before and During Total Sleep Deprivation and Recovery. Sleep 2015, 38, 1935–1943, doi:10.5665/sleep.5244.
  • Sauvet F, Erblang M , Gomez-Merino D, Rabat A, Guillard M, Dubourdieu D, Lefloch H, Drogou C, Van Beers P, Bougard C, Bourrrilhon C, Arnal P, Rein W, Mouthon F, Brunner-Ferber F, Leger D, Dauvilliers Y, Chennaoui M, Charvériat M. Efficacy of THN102 (a combination of modafinil and flecainide) on vigilance and cognition during 40-hour total sleep deprivation in healthy subjects: Glial connexins as a therapeutic target. Br J Clin Pharmacol 2019 Nov;85(11):2623-2633. doi: 10.1111/bcp.14098. Epub 2019 Sep 15.

- In Table 2, some of the “observed counts” (3rd column) do not add up to 47 participants (e.g., the second genetic polymorphism [rs1143627_IL1-β] has 8 G/G, 18 G/A and 20 A/A genotypes, adding up to N = 46 instead of 47). Please clarify. Moreover, Table 2 suggests that several participants did not only have one but several SNPs. If this is, in fact, the case, it is likely that different SNPs might have additive effects on the cognitive measures. Can the authors differentiate effects of different SNPs on these measures?  It would also be interesting to investigate whether different SNPs have additive or synergistic effects on cognitive performance.

In Results (para. 3.1), we added the information that the IL-1β genotype (and five others, ADORA2A (rs2298383 and rs5751876), ADA, DRD2, and BDNF) of one individual are missing because undetected at the probe level, although the DNA is of good quality. This is inherent to the Taqman reference method, with 1 to 4%, depending on the polymorphisms, not detected.

We agree with the notion that several participants may have not only one but several SNPs, and that different SNPs might have additive or even synergistic effects on the cognitive measures. To try to answer this we now added the statistical effects of the combination of SNPs on the evolution of PVT lapses, Go/noGo commission errors and KSS during the prolonged wakefulness.

For the 4 SNPs identified as having the largest effect on both performance tasks and KSS score during prolonged wakefulness, we conducted a combination analysis using SNPstats R software (Solé et al. 2006) to assess a potential synergistic combination of the mutations on the values observed at 26 hours of prolonged wakefulness. The results are now presented in an additional figure (Figure 6) and table (Table 4). We show the mean results for each allele combination, the mean difference and the p-value of the comparison with the most frequent combination (row 1 in the tables), using a Bonferroni test.

  • Solé, X.; Guinó, E.; Valls, J.; Iniesta, R.; Moreno, V. "SNPStats: a web tool for the analysis of association studies." Bioinformatics 22, no. 15 (2006): 1928-1929

- It would be interesting to calculate how much of the variance in the cognitive and sleepiness responses to sleep deprivation can be explained by the single SNPs as well as by combinations thereof.

According to the reviewer recommendations we added the values related to the total variance explained by the single SNP and combination. We thank the reviewer for this recommendation that improves our manuscript. 

- Participants were sleep deprived until 21:00 h of Day 3. However, cognitive measures were only taken until 6 hours earlier (15:00 h). The authors should clarify why they did not collect data until the end of sleep deprivation. Given that participants spent another night in the laboratory following sleep deprivation (i.e., a “recovery night”), it would be very interesting to see if participants’ performance on the cognitive tasks renormalized and if this differed between different genetic polymorphisms. Is it possible to provide such data?

In our study participants were sleep deprived from 7:00 at D1 until 21:00 at D2. Because of our knowledge of the cognitive responses (mainly on the PVT) to total sleep deprivation and recovery (Arnal et al., 2015, Sauvet et al., 2020), we focused on 6 time-points and the objective was to associate genetic markers of differential vulnerability. Our study was part of a doctoral thesis, with the cognitive performance on PVT being the primary outcome of interest, but secondary outcomes included the testing of physical performance. The latter was assessed in D1 and D2 from about 16:00 to 18:00 (after the PVT at 26h prolonged wakefulness), leaving little time to add a measure of cognition before dinner and bedtime. We included the recovery night and considered the end of the study at 9:00 am on day D3 for the safety of the subjects in order to avoid that they leave the laboratory in total sleep deprivation situation. We agree with the interest in future studies to see if participants’ performance after the night of recovery differed between different genetic polymorphisms.

- The authors appear to have calculated mixed-effects models which differ from an analysis of variance (ANOVA). However, they use these terms interchangeably (e.g., lines 225 ff.), leading to confusion about what statistical modeling they did use. They should clarify this in the Methods section.

According to this remark, we clarify in the Methods section. The ANOVA and posthoc results for the awakening duration effect have been added for Figure 1; effect-size analysis (Figure 2) and ANOVA analysis for the SNP effect and interaction with the awakening duration effect were performed on the nadir time point (at 26h of continuous awaking) (Table 3); we added ANOVA and posthoc results for Figures 3 to 5; the statistical effect of the four SNPs combination was added for PVT lapses, Go/noGo commission errors and KSS (Figure 6, Table 4).

- In the Results section 3.4., the outcome of the statistical analysis is unclear and only one p value is given suggesting an “awakening effect” after 26 hours of prolonged wakefulness. It is unclear what measures (PVT, Go/noGo, KSS) this test refers to and why the authors only focused on this one time point in their analyses. They should provide a more detailed analysis in this section, including main and interaction effects for time awake and performance in the different tasks. Did the authors calculate whether the results of the cognitive tasks and sleepiness differed from baseline (i.e., before sleep deprivation)?

We added ANOVA statistical analysis of the data and post-hoc for the global awakening duration effect on PVT (number of lapses and speed), Go/noGo (commission errors), and KSS score (Figure 1). As indicated above, the paragraph of statistical analysis has been clarified. As previously observed in our laboratory and publications, the nadir of performance has been observed after 26 hours of prolonged wakefulness. We clearly explained this point in the results paragraph.

- The authors used Cohen’s d as a measure of effect sizes to be reported. However, the cutoffs used for the effect sizes (Cohen’s d > 0.01 = small, > 0.056 = medium, and > 0.14 = large) do not match with the recommended cutoffs of 0.2, 0.5 and 0.8 as small, medium and large effect sizes for this measure in the literature. The authors should clarify how they chose the effect size cutoffs for their study.

In our study, the effect sizes (partial eta squared: η2p) were calculated from corresponding ANOVA F-values and degrees of freedom. Effect sizes of 0.0099, 0.0588 and 0.1379 are considered small, moderate and large, respectively (Cohen, J. Statistical power analysis for the behavioral sciences Acad. Press (2013) ; Richardson, J. T. E. Eta squared and partial eta squared as measures of effect size in educational research. Educational Research.)

  • Cohen, J. Statistical power analysis for the behavioral sciences Acad. Press (2013).
  • Richardson, J. T. E. Eta squared and partial eta squared as measures of effect size in educational research. Educational Research Review 6, 135–147 (2011).

- Why did the authors use a cutoff at > 8 for the Pittsburgh Sleep Quality Index (PSQI), although it has been suggested by the developers of this questionnaire (Buysse et al., 1989) to set the cutoff already at > 5?

We thank the reviewer; the cutoff for the PSQI is indeed 5. It was a mistake and we corrected it.

- The authors should elaborate on the difference between “predesigned” or “customized probes” participants were genotyped with (lines 213 f.). It is unclear why they used different probes for genotyping and how many participants were genotyped with one or the other. What is the difference between these probes? Are they equally well suited for this purpose?

A pre-designed probe exists in the Thermo Fisher Scientific catalog whereas for a customized probe, Thermo Fisher Scientific manufactures it on contract (custom work). In our study, the genetic analysis has been processed by the CNRGH (National center of genomic research on human) with probes well suited for the objective of our scientific collaboration.

- The authors collected actimetric data to confirm that participants stayed awake during the sleep deprivation protocol. They should indicate if this was in fact the case for all participants by providing some data. Similarly, participants were asked to fill a sleep/wake schedule for one week prior to the study. However, no results are provided concerning this measure. Did participants keep a regular sleep/wake cycle during this time?

In our study, actimetry during sleep deprivation was used to ensure that subjects believed we had an objective means of monitoring that they were not asleep; in addition, two investigators were constantly present to maintain a good level of alertness. When the subjects were about to fall asleep (eyes closed, head down), they were gently and immediately woken up (i.e., no period of sleep > 30 seconds). This has been added in Methods. In the results paragraph, the total sleep time in the table 1 has been obtained from the sleep/wake schedule completed the week before the study. This point has been added using a footnote.

- In the Results section “3.2. SNPs distribution associations”, it is unclear how the associations between SNPs with TNF-α, IL1-β, and COMT, as well as between IL1-β SNP and ADA and COMT, and TL-6 and BDNF were assessed. Please clarify.

The statistical analysis has been processed using a chi-squared test. This point has been clearly described in the statistical paragraph.

- In the Results section “3.3. Correlations between tests and questionnaire results”, the authors should provide correlation coefficients and p values for each analysis individually rather than providing summarized statistics.

We now provided correlation coefficients and p values for each analysis individually.

- Values for PVT speed are usually not normally distributed and strongly skewed. Yet, Pearson correlations were used for analyses. Please explain why using this correlation, instead of e.g., Spearman correlation, is appropriate for this data.

Overall, the data were normally distributed, regardless of wakefulness time. We then performed the Pearson test (Kolmogorov-Smirnov test for PVT speed, p = 0.065).

- In Figure 2, the authors provide effect sizes of genetic polymorphisms in the different tasks. However, the direction of these effects is unclear (i.e., whether the effect refers to a decrease or improvement in performance). It should also be added whether the effects were significant or not with respect to a specified reference.

The reviewer is wright. In the figure 2, we show the absolute value of effect size in order to compare the influence of genetic mutation. For the 4 most important SNP, we showed the direction of the effect in the figures 3, 4 and 5. We clearly described this point in the figure legend.

- While caffeine intake was prohibited 48 hours before and during the study, participants’ regular caffeine consumption was rather high at 288 ± 17 mg per day. Given that the withdrawal effect is differently affecting individuals, usually kicking in 1-2 days after stopping caffeine consumption and lasts at least several days, this might have influenced the outcome of the study, which major purpose was to investigate cognitive measures that are known to be affected by caffeine and caffeine withdrawal, possibly interacting with the effects of SNPs. Please discuss.

We agree for the remark and the possible effect of caffeine withdrawal. In our study caffeine consumption was not strictly prohibited, but we were only asking subjects not to exceed their usual consumption. This has been clearly expressed in the manuscript.

- On Day 1 of the 3 days in the laboratory, total sleep deprivation began at 7:00 AM. Was this time matched with participants preferred wakeup time (e.g., based on the score in the morningness-eveningness questionnaire or the one-week sleep schedule)?

The subjects were mainly of intermediate or morningness chronotype ; moreover, the 7:00 a.m. schedule is the most representative of the schedule of professionally active people.

- It is unclear why only some SNPs are shown in Figures 3 – 5. In addition, the authors should provide detailed statistical analyses of the data shown in these figures. How do the SNPs of interest compare to each other and with respect to the tertiles?

We thank the reviewer for this suggestion, and we now added the posthoc results in the figures 2 – 5.

- The language is sometimes difficult to understand. E.g., what do the authors mean with “constant causality relation” (e.g., line 282 or lines 313 f.) and “… target polymorphisms that are immediately involved in vulnerability to sleep loss” (lines 130 f.)?

We apologized as these expression were not appropriate. We have changed the « constant causality relation » line 282 for « The ANOVA statistical analysis evidenced a significant main awakening effect with the decrease of sustained attention in the PVT…”. In the line 313, the results were clarified. We have changed « target » for “The second objective is to evaluate influences of genetic polymorphisms that are immediately involved in vulnerability to sleep loss ».

- Line 242: What do the authors mean by “controlling for study”?

This was not appropriate, and we deleted it. "

Minor issues

- The participants’ age in Table 1 (37.0 ± 1.1 years) does not match the number reported in the Abstract and Methods (38.2 ± 2.1 years). Please correct.

We apologized for the error, and we corrected for : 37.0 ± 1.1 years as indicated in the Table 1.

- The naming of the repeated-measures factor “awakening time” (e.g., line 227) is misleading because it implies that participants were woken up during these times. I suggest the authors to rename this factor to “time awake” or similar to avoid confusion.

We thank the reviewer and we corrected for « time awake ».

- What do the authors mean by “approximately” after 2, 8, 14 hours etc. (line 171)? How much deviation was there between individual participants?

In our study, the evaluation during the whole protocol of sleep deprivation has been performed in sessions of four by four subjects. Thus the deviation was minimal, at most 5 min; and subjects started the PVT and the Go/noGo exactly together. The term « approximatively » has been suppressed.

- I found it untypical to report degrees of freedom for the individual analyses in the Methods section. The authors should consider moving these measures to the respective results parts. It is also not clear why the degrees of freedom are different for the different measures. Please specify.

The degrees of freedom for the individual analyses have been moved to the respective results part. The degrees of freedom for the sleep deprivation effect is F(5,42) because 6 time points of awaking and 47 subjects; the degrees of freedom for the genotype effect is F(2,46) if 3 genotypes and 47 subjects or F(1,46) if 2 genotypes (heterozygote + homozygote mutated) and 47 subjects.

- What was the reasoning behind the authors’ decision to only include participants who scored between 31 and 69 in the morningness-eveningness questionnaire?

This was to avoid taking extreme subjects in each of the chronotypes.

- There are several typos in the manuscript.

E.g., abstract: the acronym “SDT” should probably read “TSD” (total sleep deprivation)?

Abstract (line 64): should read “ADORA2A”.

Lines 200 f.: should read “Subjects have 2 s to respond…”.

Lines 217 f.: should read “… sequencing of PCR products was carried out for…”.

Line 232: should read “… and for polymorphism effect (SNP effect) are respectively …”.

Line 233: should read “F(2,46) or F(1,46).”.

Line 258: should read “Therefore, if group of homozygous mutations counts…”.

Title of Table 2 (line 277): should read “… in the data base of 1000 genomes”

We thank the reviewer, and we apologized for the typos. We now corrected adequately.

Reviewer 2 Report

First, I would like to congratulate the authors with a highly interesting paper that is well written and contributes to the already existing literature. A lot of effort was clearly put into the design and the execution of the study for which I would certainly like to congratulate the authors.

Abstract:

  • Line 62: please be consistent in your abbreviations. In line 55 you define total sleep deprivation as TSD in Line 62 you use SDT.

Introductions:

  • Line 79: please change increased to increases and decreased to decreases.
  • Line 94: have shown an association or have shown associations
  • Line 97: has also been demonstrated
  • Line 97: I think you mean to say ‘…demonstrated to play a role in the occurrence of inter-individual differences in vulnerability to neurobehavioral impairments during sleep loss.’ I don’t think inter-individual vulnerability is the correct way of saying.

From this point on I’m going to stop checking grammar, I think it is clear that a grammatical check of this paper is appropriate to promote readability.

  • Line 98-101: In my opinion these sentences do not substantiate a role of TNF-alpha in the existence of inter-individual differences in vulnerability to neurobehavioral impairments during sleep loss. Please revise/elaborate.
  • Line 113: please insert a closing bracket.
  • Line 89-118: by reading this paragraph it is clear that different genetic polymorphisms appear to play a role in different aspects of neurobehavioral performance. Please elaborate on this, do these differences occur due to methodological differences between studies? Or are there other reasons that might play a role?
  • Line 119-122: I think you make an important point here that deserves more attention in your introduction. A more profound introduction in the different available techniques to assess genetic polymorphisms would improve your introduction. As such you can elaborate more on the discussion why the deleterious effects of TSD on neurobehavioral performances would not be based on a single genetic polymorphism, and you could also substantiate why you eventually decided to focus on 14 SNPs and indicate how this ‘method’ is superior to only focus on 1 SNP.
  • Line 124-125: Why did you decide to specifically focus on sustained attention, executive function and subjective sleepiness? Is this because these are important features in the occupations that you mentioned in the beginning of your introduction? Please elaborate.
  • Line 126: Why don’t you mention the BDNF-polymorphism in the second paragraph of your introduction as well?
  • Line 130: please change ‘is’ to ‘was’

Methods:

  • Line 200: Please delete ‘to’ after ‘subjects have’
  • Line 233: Please change ‘ou’ to ‘or’

Results:

  • Line 273: Please change ‘lesser’ to ‘less’
  • Line 320: Please change ‘represented’ to ‘represents’
  • Section 2.6: You mention a couple of significant interaction effects, but you did not follow these up with independent-samples t-tests to interpret this interaction effect. Is there a specific reason? In my opinion, the results of these t-tests would be of added value and certainly when they are also incorporated in figure 3, 4 and 5 (which are really clear already btw!)
  • Overall results: An analysis in which you group participants not only per individual SNP, but also per a specific combination of multiple SNPs (e.g. you group participants in most vulnerable and least vulnerable, based on the combination of all SNPs) would be of added value. This would also emphasize the importance of not solely focusing on individual SNPs, but indicate that it is important to take into account multiple SNPs at the same time in order to predict responses on a behavioral level. These results could also be of added value in the discussion.

Discussion:

  • Line 521: Two times ‘is an enzyme’
  • Line 526: Please change ‘reduces’ to ‘reduce’
  • Line 527: Please change ‘in’ to ‘is’
  • Overall discussion: Overall I enjoyed the in depth discussion of the specific role of each SNP in the vulnerability to sleep loss. However, before going into this discussion I think it would be valuable to first discuss the effect of sleep loss on neurobehavioral performance in general. PVT lapses and Go/noGo commission errors deteriorated, but PVT speed improved with prolonged wakefulness. I’m of the opinion these findings also deserve to be discussed in your discussion.

Conclusion:

  • You did not mention anything about the transferability of these results to a military situation in discussion. In addition, you also did not mention anything about banking sleep in limiting the mentioned TSD-associated vigilance impairments in discussion. Please do so before incorporating this in your conclusion.

Author Response

First, I would like to congratulate the authors with a highly interesting paper that is well written and contributes to the already existing literature. A lot of effort was clearly put into the design and the execution of the study for which I would certainly like to congratulate the authors.

We thank the reviewer.

Abstract:

  • Line 62: please be consistent in your abbreviations. In line 55 you define total sleep deprivation as TSD in Line 62 you use SDT.

We apologize for the mistake, and we corrected adequately.

Introductions:

  • Line 79: please change increased to increases and decreased to decreases.
  • Line 94: have shown an association or have shown associations
  • Line 97: has also been demonstrated
  • Line 97: I think you mean to say ‘…demonstrated to play a role in the occurrence of inter-individual differences in vulnerability to neurobehavioral impairments during sleep loss.’ I don’t think inter-individual vulnerability is the correct way of saying.

We agree on this accuracy and we have corrected the sentence in this sense.

From this point on I’m going to stop checking grammar, I think it is clear that a grammatical check of this paper is appropriate to promote readability.

  • Line 98-101: In my opinion these sentences do not substantiate a role of TNF-alpha in the existence of inter-individual differences in vulnerability to neurobehavioral impairments during sleep loss. Please revise/elaborate.

We agree with the reviewer remark, and we revise this sentence as follow : « This was based on previous studies showing the role of TNF-α in sleep/wake regulation (Krueger et al., 2010). The TNFα polymorphism (rs1800629) has been found associated with phenotypic inter-individual differences in vulnerability to PVT performance impairment related to sleep loss (Satterfield et al., 2015), and common symptoms experienced across chronic conditions such as pain, sleep, fatigue, and cognitive symptoms (Knisely et al., 2019). Interestingly, this functional polymorphisms impact either the secretion of the protein or the level of transcription of the gene (Louis et al., 1998 ; Wilson et al., 1997).

  • Krueger, JM. The Role of Cytokines in Sleep Regulation. Curr Pharm Des 2008, 14, 2008;14(32):3408-16. doi: 10.2174/138161208786549281.
  • Satterfield, B.C.; Wisor, J.P.; Field, S.A.; Schmidt, M.A.; Van Dongen, H.P.A. TNFα G308A Polymorphism Is 613 Associated with Resilience to Sleep Deprivation-Induced Psychomotor Vigilance Performance Impairment in 614 Healthy Young Adults. Brain Behav. Immun. 2015, 47, 66–74, doi:10.1016/j.bbi.2014.12.009. 615
  • Knisely MR , Maserati M, Heinsberg LW, Shah LL , Li  H, Zhu Y , Ma Y , Graves LY, Merriman JD , Conley YP. Symptom Science: Advocating for Inclusion of Functional Genetic Polymorphisms. Biol Res Nurs 2019 Jul;21(4):349-354. doi: 10.1177/1099800419846407. Epub 2019 Apr 25.
  • Louis, E.; Franchimont, D.; Piron, A.; Gevaert, Y.; Schaaf-Lafontaine, N.; Roland, S.; Mahieu, P.; Malaise, M.; De Groote, D.; Belaiche, J. Tumour necrosis factor (TNF) gene polymorphism influences TNF-alpha production in lipopolysaccharide (LPS)-stimulated whole blood cell culture in healthy humans. Clin. Exp. Immunol. 1998, 113, 401–406.
  • Wilson, A.G.; Symons, J.A.; McDowell, T.L.; McDevitt, H.O.; Duff, G.W. Effects of a polymorphism in the human tumor necrosis factor promoter on transcriptional activation. Natl. Acad. Sci. USA 1997, 94, 3195–3199.

  • Line 113: please insert a closing bracket.

This has been done.

  • Line 89-118: by reading this paragraph it is clear that different genetic polymorphisms appear to play a role in different aspects of neurobehavioral performance. Please elaborate on this, do these differences occur due to methodological differences between studies? Or are there other reasons that might play a role?

The differences in the influence of genetic polymorphism (ADORA2A, COMT, DAT1, DRD2) on cognitive vulnerability to TSD are related to their involvement in different levels of cognition. We added at line 85:

 “The influence of single nucleotide polymorphisms (SNPs) on cognitive responses to TSD has been mainly observed on tasks involved in different levels of cognition, such as the psychomotor vigilance test (PVT) and the executive Go/noGo inhibition task, and also working memory, decision making, and flexibility [8]. The PVT is one of the most sensitive cognitive tests to sleep deprivation-related sleep pressure, which induces increased levels of extracellular adenosine and increased neuronal activity in a parieto-frontal network (Porkka-Heiskanen et al., 2000). The core executive functioning (such as behavioral inhibition, working memory, and cognitive flexibility) requests optimal func-tioning of parieto-frontal and fronto-striatal networks, served by COMT activity, the gene of catechol-O-methyl transferase, the main enzyme degrading catecholamines; and dopamine levels (Logue and Gould, 2014).”

  • Porkka-Heiskanen T, Strecker RE, McCarley RW. Brain site-specificity of extracellular adenosine concentration changes during sleep deprivation and sp RWontaneous sleep: an in vivo microdialysis study Neuroscience 2000;99(3):507-17. doi: 10.1016/s0306-4522(00)00220-7.
  • Logue SF, Gould The neural and genetic basis of executive function: attention, cognitive flexibility, and response inhibition. Pharmacol Biochem Behav. 2014 Aug;123:45-54. doi: 10.1016/j.pbb.2013.08.007. Epub 2013 Aug 24.

  • Line 119-122: I think you make an important point here that deserves more attention in your introduction. A more profound introduction in the different available techniques to assess genetic polymorphisms would improve your introduction. As such you can elaborate more on the discussion why the deleterious effects of TSD on neurobehavioral performances would not be based on a single genetic polymorphism, and you could also substantiate why you eventually decided to focus on 14 SNPs and indicate how this ‘method’ is superior to only focus on 1 SNP.

We agree the reviewer and we now added this paragraph:

“Among available techniques to assess genetic polymorphism, the sequencing and the conventional TaqMan PCR methods are highly reliable, but sequencing is very expensive (Komar 2009). There is also the Sequenom MassARRAY iPLEX Platform, based on distinguishing allele-specific primer extension products by mass spectrometry (MALDI-TOF), but conditioned by the molecular weights of alternative extension products.  At last, our lab has evaluated the LAMP-MC (Loop-mediated isothermal amplification and melting curve Analysis) method on blood and buccal cells, without prior DNA extraction, for detection of small SNPs sample size (n=5) (Drogou et al., 2020). Another technique is the array-based hybridization, which consisted in chips where specific probes of a panel of genes are attached. Throughout our study, a research agreement between IRBA and CNRGH (National research center in human genomics – François Jacob Biological Institute, CEA, Evry, France) was previously established to perform the conventional Taqman method for studying the influence of fourteen candidate genes on self-reported sleep characteristics and caffeine consumption in a French working population of workers (n = 1023 participants (Erblang et al., 2019).”

  • Komar AA. Single Nucleotide Polymorphisms - Methods and Protocols. Springer Protocols, 2009, Second edition, Edited by Anton A. Komar. DOI https://doi.org/10.1007/978-1-60327-411-1
  • Drogou, C.; Sauvet, F.; Erblang, M.; Detemmerman, L.; Derbois, C.; Erkel, M.C.; Boland, A.; Deleuze, J.F.; Gomez-Merino, D.; Chennaoui, M. Genotyping on blood and buccal cells using loop-mediated isothermal amplification in healthy humans. Biotechnol. Rep. 2020, 26, e00468.
  • Erblang, M.; Drogou, C.; Gomez-Merino, D.; Metlaine, A.; Boland, A.; Deleuze, J.F.; Thomas, C.; Sauvet, F.; 598 Chennaoui, M. The Impact of Genetic Variations in ADORA2A in the Association between Caffeine Consumption 599 and Sleep. Genes 2019, 10, 1021, doi:10.3390/genes10121021.

  • Line 124-125: Why did you decide to specifically focus on sustained attention, executive function and subjective sleepiness? Is this because these are important features in the occupations that you mentioned in the beginning of your introduction? Please elaborate.

We focus on sustained attention, executive function and subjective sleepiness because it is a continuation of our works on the cognitive consequences of total sleep deprivation or sleep restriction, and the interest of countermeasures (sleep extension, nap, blue-enriched light, caffeine, exercise) (Arnal et al., 2015, Sauvet et al., 2015, 2020, Faraut et al., 2015, 2020). This is particularly important for military personnel (pilots, infantry, sailors, who are often in external operations - OPEX) who represent the target population of our professional institution. In our works, although our subjects are rather young (about 30 years old) and healthy, we also observed inter-individual differences in vulnerability to neurobehavioral impairments during total sleep deprivation, and we were interested in the influence of genetic polymorphisms using a candidate gene approach.

We added in the first paragraph “Sustained attention, executive function and subjective sleepiness are particularly im-portant for military personnel (pilots, infantry, sailors, who are often on external opera-tions - OPEX).”

  • Arnal, P.J.; Sauvet, F.; Leger, D.; van Beers, P.; Bayon, V.; Bougard, C.; Rabat, A.; Millet, G.Y.; Chennaoui, M. Benefits of Sleep Extension on Sustained Attention and Sleep Pressure Before and During Total Sleep Deprivation and Recovery. Sleep 2015, 38, 1935–1943, doi:10.5665/sleep.5244.
  • Sauvet, F.; Arnal, P.J.; Tardo-Dino, P.-E.; Drogou, C.; Van Beers, P.; Erblang, M.; Guillard, M.; Rabat, A.; Malgoyre, A.; Bourrilhon, C.; Léger, D.; Gomez-Mérino, D.; Chennaoui, M. Beneficial Effects of Exercise Training on Cognitive Performances during Total Sleep Deprivation in Healthy Subjects. Sleep Med. 2020, 65, 26–35.
  • Sauvet, F.; Gomez, D.. Rabat, A.; Chennaoui, M. Guide pratique. Gestion du cycle veille - sommeil en milieu militaire. EAN: 9782914558938; 2020. 83 p.
  • Faraut B, Léger D, Medkour T, Dubois A, Bayon V, Chennaoui M, Perrot S. Napping reverses increased pain sensitivity due to sleep restriction PLoS One 2015 Feb 27;10(2):e0117425. doi: 10.1371/journal.pone.0117425. eCollection 2015.
  • Faraut B, Andrillon T, Drogou C, Gauriau C, Dubois A, Servonnet A, Van Beers P, Guillard M, Gomez-Merino D, Sauvet F, Chennaoui M, Léger D. Daytime Exposure to Blue-Enriched Light Counters the Effects of Sleep Restriction on Cortisol, Testosterone, Alpha-Amylase and Executive Processes. Front Neurosci 2020 Jan 8;13:1366. doi: 10.3389/fnins.2019.01366. eCollection 2019.
  • Line 126: Why don’t you mention the BDNF-polymorphism in the second paragraph of your introduction as well?

We now added the interest of the BDNF polymorphism as follow:

“The BDNF Val66Met polymorphism (rs6265) has been found to modulate sleep intensity and to impact an individual’s vulnerability to sleep deprivation (Bachman et al., 2012, Grant et al., 2018), with Met carriers performing more poorly on neurobehavioral tasks than Val/Val individuals during extended wakefulness. The neurotrophic BDNF is an established mediator of long-term synaptic plasticity, neurogenesis and learning and memory (Cunha et al., 2010).”

  • Bachmann, V.; Klein, C.; Bodenmann, S.; Schäfer, N.; Berger,W.; Brugger, P.; Landolt, H.P. The BDNF Val66Met polymorphism modulates sleep intensity: EEG frequency- and state-specificity. Sleep 2012, 35, 335–344.
  • Grant, L.K.; Cain, S.W.; Chang, A.-M.; Saxena, R.; Czeisler, C.A.; Anderson, C. Impaired Cognitive Flexibility during Sleep Deprivation among Carriers of the Brain Derived Neurotrophic Factor ( BDNF ) Val66Met Allele. Behav. Brain Res. 2018, 338, 51–55, doi:10.1016/j.bbr.2017.09.025
  • Cunha C, Brambilla R, Thomas KL. A simple role for BDNF in learning and memory? Front Mol Neurosci 2010 Feb 9;3:1. doi: 10.3389/neuro.02.001.2010. eCollection 2010.

  • Line 130: please change ‘is’ to ‘was’

This was done.

Methods:

  • Line 200: Please delete ‘to’ after ‘subjects have’
  • Line 233: Please change ‘ou’ to ‘or’

We thank the reviewer, and we changed appropriately for both remarks.

Results:

  • Line 273: Please change ‘lesser’ to ‘less’

This has been changed.

  • Line 320: Please change ‘represented’ to ‘represents’

This was done.

  • Section 2.6: You mention a couple of significant interaction effects, but you did not follow these up with independent-samples t-tests to interpret this interaction effect. Is there a specific reason? In my opinion, the results of these t-tests would be of added value and certainly when they are also incorporated in figure 3, 4 and 5 (which are really clear already btw!)

We thanks the reviewer for this review. We added the post hoc analysis in figures 3, 4 and 5

  • Overall results: An analysis in which you group participants not only per individual SNP, but also per a specific combination of multiple SNPs (e.g. you group participants in most vulnerable and least vulnerable, based on the combination of all SNPs) would be of added value. This would also emphasize the importance of not solely focusing on individual SNPs, but indicate that it is important to take into account multiple SNPs at the same time in order to predict responses on a behavioral level. These results could also be of added value in the discussion.

We thank the reviewer for this suggestion, and we now added the combination analysis for the four respective SNPs illustrated in figures 3 to 5 (PVT lapses, Go/noGo commission errors and KSS score). We added a 2.7 paragraph on the SNPs combination analysis and results on Table 4 and Figure 6. We also added the interest of the combination in the discussion.

Discussion:

  • Line 521: Two times ‘is an enzyme’
  • Line 526: Please change ‘reduces’ to ‘reduce’
  • Line 527: Please change ‘in’ to ‘is’

We thank the reviewer, and we have corrected appropriately for all three remarks.

  • Overall discussion: Overall I enjoyed the in depth discussion of the specific role of each SNP in the vulnerability to sleep loss. However, before going into this discussion I think it would be valuable to first discuss the effect of sleep loss on neurobehavioral performance in general. PVT lapses and Go/noGo commission errors deteriorated, but PVT speed improved with prolonged wakefulness. I’m of the opinion these findings also deserve to be discussed in your discussion.

We thank the reviewer for this remark. As suggested, we now first discussed the overall effect of sleep deprivation on neurobehavioral performance as follow:

“We first illustrated in Figure 1 the significant awakening duration effect and individual variability on cognitive responses and subjective sleepiness, with a significant increase in the number of PVT lapses and a decrease in speed, as well as an increase in commission errors in the Go/noGo inhibition executive task.”

We do not understand why the reviewer refers to an improvement in PVT speed (corresponding to “1 / reaction time × 1000”), since the figure 1 shows its decrease. Therefore, we changed the Y-axis legend to “1/RT X 1000”. Our results thus illustrated the overall decrease of PVT (lapses and speed) and Go/noGo commission errors during total sleep deprivation.

.

Conclusion:

  • You did not mention anything about the transferability of these results to a military situation in discussion. In addition, you also did not mention anything about banking sleep in limiting the mentioned TSD-associated vigilance impairments in discussion. Please do so before incorporating this in your conclusion.

We added elements on the transferability of our results to the military population, and we also provided a reminder of the effects of banking sleep to limit performance impairments related to total sleep deprivation ; as follow:

“…These results could be used for the benefit of the Forces to individualize and optimize countermeasure strategies to limit performance impairments induced by sleep deprivation. In a previous study, Maire et al. [23] evidenced that manipulation of the sleep homeostatic state affects sustained attention and sleepiness differentially based on PER3-dependent vulnerability. As an example, we recently showed that six nights of extended sleep (approximately 1.2 h/night) protects against vigilance impairments during TSD and improves their recovery speed (Arnal et al., 2015), which would deserve to be studied under the influence of the three genetic determinants."

Round 2

Reviewer 1 Report

The authors addressed all comments to my satisfaction.

Reviewer 2 Report

All my remarks were resolved. Thank you for these revisions.